# Bacteriophage infection drives loss of β-lactam resistance in methicillin-resistant *Staphylococcus aureus*

**My Tran[†], Angel J Hernandez Viera[†], Patricia Q Tran, Erick D Nilsen, Lily Tran, Charlie Y Mo***

Department of Bacteriology, University of Wisconsin-Madison, Madison, United States

## eLife Assessment

The manuscript explores how bacterial evolution in the presence of lytic phages modulates b-lactams resistance and virulence properties in methicillin-resistant *Staphylococcus aureus* (MRSA). This **important** work improves our knowledge of how mutation in genes required for phage infection confers sensitivity to b-lactams and alter virulence properties. Altogether, the findings are **convincing**.

**\*For correspondence:**
cymo@wisc.edu

[†]These authors contributed equally to this work

**Abstract** Bacteriophage (phage) therapy is a promising means to combat drug-resistant bacterial pathogens. Infection by phage can select for mutations in bacterial populations that confer resistance against phage infection. However, resistance against phage can yield evolutionary trade-offs of biomedical relevance. Here, we report the discovery that infection by certain staphylococcal phages sensitizes different strains of methicillin-resistant *Staphylococcus aureus* (MRSA) to β-lactams, a class of antibiotics against which MRSA is typically resistant. MRSA cells that survive infection by these phages display significant reductions in minimal inhibitory concentration against different β-lactams compared to uninfected bacteria. Transcriptomic profiling reveals that these evolved MRSA strains possess highly modulated transcriptional profiles, where numerous genes involved in *S. aureus* virulence are downregulated. Phage-treated MRSA exhibited attenuated virulence phenotypes in the form of reduced hemolysis and clumping. Despite sharing similar phenotypes, whole-sequencing analysis revealed that the different MRSA strains evolved unique genetic profiles during infection. These results suggest complex evolutionary trajectories in MRSA during phage predation and open up new possibilities to reduce drug resistance and virulence in MRSA infections.

## Introduction

*Staphylococcus aureus* is one of the most notorious and widespread bacterial pathogens, responsible for hundreds of thousands of severe infections worldwide every year. Methicillin-resistant *S. aureus* (MRSA) poses a particular clinical threat, as MRSA infections increase mortality, morbidity, and hospital stay, as compared to those caused by methicillin-sensitive *S. aureus* (MSSA; *Ippolito et al., 2010*). Part of MRSA's notoriety stems from its strong resistance against the β-lactam family of antibiotics, such as penicillins and cephalosporins, which inhibit the activity of transpeptidase enzymes during peptidoglycan synthesis in bacterial cell walls (*Tomasz, 1979*). β-lactams are one of the most commonly prescribed drug classes, with many designated as 'Critically Important' antimicrobials by the World Health Organization (*WHO, 2019*). Thus, MRSA infections pose a considerable public health risk as they are notoriously difficult to treat and are widespread in communities and hospital settings. Indeed,

**eLife digest** MRSA (short for methicillin-resistant *Staphylococcus aureus*) is a commonly found group of bacteria that spread primarily through skin-to-skin contact. Unlike other *S. aureus* strains, MRSA is resistant to a class of commonly used antibiotics known as ß-lactams. As a result, MRSA infections are extremely difficult to treat and can lead to potentially life-threatening conditions, such as pneumonia and sepsis.

It has been suggested that another way to eliminate drug-resistant bacteria like MRSA is to treat them with bacteriophages, viruses that specifically infect and kill bacteria. However, bacteria can become resistant to these phages as well. Interestingly, this can result in an evolutionary trade-off: as bacteria become more resistant to phages, they become less resistant to antibiotics. Could this trade-off also occur in MRSA?

To investigate, Tran et al. exposed different strains of MRSA to bacteriophages. While most bacterial cells were killed off, a small fraction survived and regrew to form a new population. Tran et al. found that the phage-resistant MRSA population became sensitive once again to ß-lactam antibiotics. Genetic analysis also revealed that the regrown MRSA population activated a different set of genes. Specifically, they downregulated genes that trigger and promote infections, as well as genes associated with cell-to-cell communication.

These findings suggest that bacteriophages could be a valuable tool for restoring antibiotic sensitivity in MRSA and offer fresh insights into how drug resistance evolves. Additionally, the study highlights how *S. aureus* bacteria genetically respond to bacteriophage infections. Further research is needed to better understand the molecular mechanisms behind this phage- and antibiotic-resistance trade-off.

in 2019, MRSA alone accounted for more than 100,000 deaths attributable to drug-resistant infections worldwide (*Murray et al., 2022*).

A chief mediator of β-lactams resistance in MRSA is the SCCmec cassette, a mobile genetic element that carries the resistance gene *mecA*. MecA encodes for the penicillin-binding protein 2 A (PBP2a), a transpeptidase that has a low affinity for β-lactams (*Zapun et al., 2008*). This lower affinity permits PBP2a to participate in peptidoglycan synthesis even in the presence of β-lactams, ultimately resulting in cell survival. In addition to *mecA*, numerous MRSA strains also carry β-lactamases, such as BlaZ, that degrade β-lactams, thus further contributing to drug resistance. Together, these mechanisms can severely limit treatment options against MRSA, with current clinical treatment options relying primarily on vancomycin and daptomycin (*Liu et al., 2011*). Both vancomycin and daptomycin are last resort antibiotics against MRSA, and a major concern is the increasing resistance of MRSA to these drugs (*Turner et al., 2019*; *Barros et al., 2019*). Developing solutions to combat MRSA is a major focus in academia and industry.

Due to its drug resistance and clinical burden, *S. aureus* is a prime candidate for alternative antimicrobial treatments, such as bacteriophage (phage) therapy. Phages are viruses that infect and kill bacteria, posing one of the greatest existential threats to bacterial communities, with some estimates suggesting that 40% of all bacterial mortality worldwide is caused by phage predation (*Wilhelm and Suttle, 1999*). In phage therapy, lytic phages are administered to kill the bacterial pathogen(s) causing an infection. Phages offer certain advantages over traditional antibiotics: they are highly specific to their hosts by reducing off-target killing; they self-amplify and evolve, enabling the rapid generation of new phage variants with improved activities; and they are generally regarded as safe, as toxicity has been reported only in extremely rare cases in animals and patients (*Kortright et al., 2019*). Indeed, against *S. aureus* infections, over a dozen promising case studies and clinical trials have been reported in the past decade (*Hatoum-Aslan, 2021*).

Despite these advances, routine use of phage therapy is still met with challenges. Chief among these is the inevitable rise of phage resistance, as phage predation exerts a strong selective pressure on bacterial populations. According to one meta-analysis that focused on phage therapy outcomes, resistance against phage evolved in 75% of human clinical cases in which the evolution of resistance was monitored (*Oechslin, 2018*). Mutations represent a chief pathway by which bacteria evolve resistance against phage. To date, the best characterized phage resistance mutations involve alterations

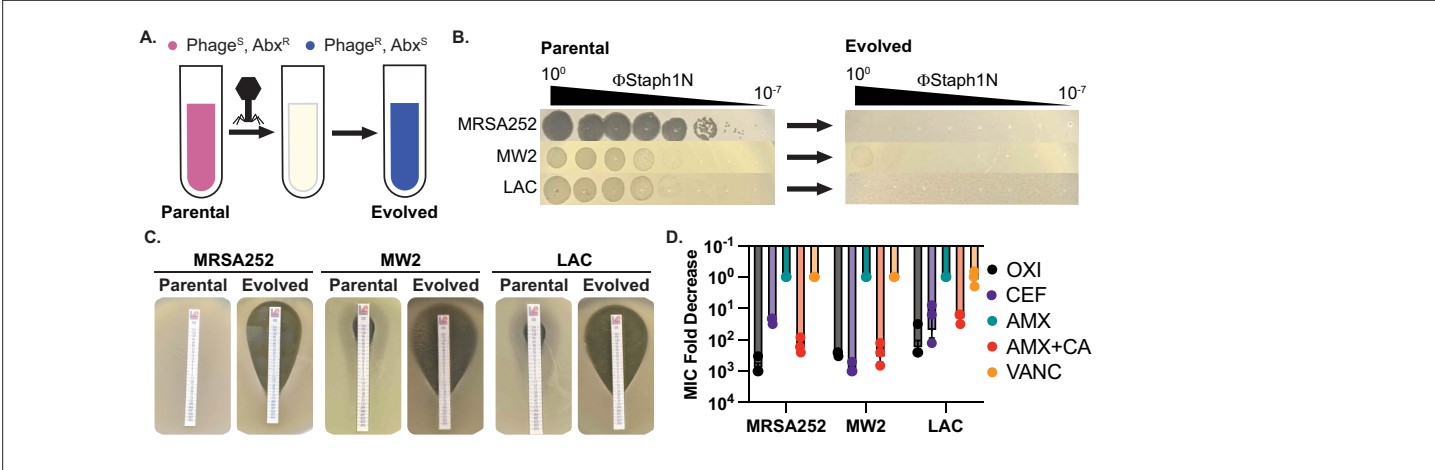

**Figure 1.** Infection by bacteriophage ΦStaph1N drives the loss of β-lactam resistance in MRSA. (**A**) Schematic of the experimental setup. Drug-resistant (AbxR), phage-sensitive (PhageS) bacterial cultures are infected with phage. The population of infected cells is passaged and allowed to recover. The surviving cell population is resistant to phage infection (PhageR) but has evolved sensitivity to antibiotics (AbxS). (**B**) ΦStaph1N infects MRSA strains MRSA252, MW2, and LAC (left panel). Following infection with ΦStaph1N, evolved cultures of the three MRSA strains are resistant to ΦStaph1N (right panel). (**C**) ΦStaph1N-treated, evolved MRSA strains show significant loss of resistance against oxacillin, compared to the parental strains. Loss of resistance is indicated by the area of bacterial clearance surrounding the antibiotic resistance strip. (**D**) ΦStaph1N treatment causes loss of resistance against different β-lactams. Plotted are the fold reductions of minimal inhibitory concentration (MIC) between treated and mock-treated cells. OXI = oxacillin; CEF = cefazolin; AMX = amoxicillin; AMX + CA = amoxicillin and clavulanic acid; VANC = vancomycin. Error bars represent the Standard Error of the Mean (SEM) of three independent replicates.

The online version of this article includes the following source data and figure supplement(s) for figure 1:

**Source data 1.** Uncropped plate images for *Figure 1B and C*.

**Source data 2.** Source data for the bar graphs in *Figure 1D*.

**Figure supplement 1.** Phage sensitivity of MRSA strains.

**Figure supplement 1—source data 1.** Uncropped plate images of *Figure 1—figure supplement 1*.

**Figure supplement 2.** Growth curves of MRSA strains under varying levels of ΦStaph1N infection.

on cell surface receptor molecules that mediate phage attachment. In many bacteria, these receptors are often proteins or sugar moieties, which are recognized by phage proteins (*Hatoum-Aslan, 2021*; *Azam and Tanji, 2019*; *Gerlach et al., 2018*). For example, in *Escherichia coli*, mutations in the cell-wall protein LamB confer resistance against *lambda* phage infection, while in *S. aureus*, mutations that modify wall teichoic acid (WTA) have been shown to limit phage infection (*Gerlach et al., 2018*; *Meyer et al., 2012*). Complicating the picture, studies have revealed that a plethora of additional host mechanisms, including dedicated anti-phage defense systems, can impact the evolution of resistance against phage (*Bernheim and Sorek, 2020*; *Maynard et al., 2010*). These problems highlight the importance of developing phage treatment strategies that minimize or capitalize on the evolution of phage resistance (*Oromí-Bosch et al., 2023*).

A unique aspect of phage therapy is the possibility to exploit evolutionary trade-offs to combat resistant pathogens. A genetic trade-off is defined as an evolved trait that confers a fitness advantage against a particular selective pressure at the expense of reduced fitness against an unselected pressure. Across many different species of bacteria, such trade-offs have been shown to occur between phage resistance and antibiotic resistance (*Figure 1A*). Phages that bind to a virulence factor or mechanism for antibiotic resistance in the target bacteria are predicted to exert a strong selection pressure on the bacteria to mutate or downregulate the phage-binding target. These changes would confer protection against phage infection but could in turn reduce the resistance or virulence in the bacterium. As an example, in *P. aeruginosa,* infection by the phage OMKO1, which binds to the outer membrane protein M OprM of MexAB- and MexXY-OprM efflux pumps, drives the evolution of mutations in those genes, leading to the re-sensitization of phage-resistant *P. aeruginosa* mutants to antibiotics (*Kortright et al., 2019*; *Chan et al., 2016*).

Little is known about how phage resistance can mediate genetic trade-offs in MRSA. Previous work has shown that phage resistance in *S. aureus* can proceed through genetic mutations that are not directly involved in phage binding. For example, studies by Berryhill and colleagues demonstrated that phage infection of *S. aureus* Newman, an MSSA strain, can select for mutations in *femA*, which is a cytoplasmic enzyme that catalyzes the formation of the pentaglycine bridge of peptidoglycans in *S. aureus* (*Berryhill et al., 2021*). Rather than serving as a direct phage receptor molecule, *femA* maintains the integrity of the cell wall, which in turn could be vital for WTA maturation and phage attachment (*Azam and Tanji, 2019*; *Xia and Wolz, 2014*). Interestingly, a consequence of these *femA* mutations is increased sensitivity against antibiotics. We therefore asked how phage resistance might impact the physiology of drug-resistant MRSA, with the hopes of identifying genetic trade-offs of potential biomedical relevance.

In the work reported here, we show that infection by staphylococcal phages causes MRSA strains to evolve sensitivity to different types of β-lactam antibiotics and attenuate virulence phenotypes. We found that this loss of resistance and virulence is associated with distinct mutational profiles distinct in each MRSA strain, and that phage-treated, evolved MRSA populations display significant transcriptome remodeling. Unexpectedly, we also discovered a mutant phage with higher activity and a broader host range against MRSA. Findings from our work can help in the development of phage therapies that reduce drug resistance and virulence in pathogenic bacteria.

## Results

### Identification of ΦStaph1N with activity against multiple MRSA strains

For our studies, we focused on three MRSA strains – MRSA252 (USA200), MW2 (USA400), and LAC (USA300). All three MRSA strains are pathogenic isolates implicated in human disease and are used as representative examples for studying MRSA (*Voyich et al., 2005*; *Holden et al., 2004*; *Kobayashi et al., 2011*). To test for phage susceptibility, we performed plaquing assays with a panel of staphylococcal phages (*Figure 1*; *Figure 1—figure supplement 1*). Of the phages tested, one phage, ΦStaph1N, which belongs to the *Kayvirus* genus, formed plaques on all three MRSA strains (*Figure 1B*; *Xia and Wolz, 2014*; *Deghorain and Van Melderen, 2012*). Yet, despite its ability to infect all three strains, ΦStaph1N infection was unable to eradicate MRSA cultures. Both MW2 and LAC displayed incomplete lysis in liquid culture at multiplicities of infection (MOIs) of 0.1 or lower (*Figure 1—figure supplement 2*). Furthermore, infected cultures of all three MRSA strains could recover back to high cell density after passaging one percent of the culture into fresh media following 24 hr of initial infection. These results suggest that infection by ΦStaph1N selects for resistant mutants that could sweep the population. Indeed, ΦStaph1N was unable to form plaques on recovered MRSA cultures that survived in the initial ΦStaph1N infection (*Figure 1B*).

### Resistance against ΦStaph1N infection sensitizes MRSA against β-lactams

Because both phages and β-lactams interface with the bacterial cell wall, we hypothesized that resistance against ΦStaph1N infection could cause a trade-off in β-lactam resistance in MRSA even in the presence of PBP2a and BlaZ. We first tested the β-lactam sensitivity of the parental MRSA252, MW2, and LAC strains. As expected, all three strains displayed high minimal inhibitory concentrations (MICs) of ≥48 μg/mL against the β-lactams oxacillin (OXA), cefazolin (CEF), amoxicillin (AMX), and amoxicillin and clavulanic acid (AMX + CA), visually indicated by their ability to form lawns surrounding antibiotic strips (*Figure 1C*). The strains were sensitive to vancomycin (VANC; MICs = 1.5 μg/mL), which inhibits cell wall synthesis through a different mechanism than β-lactams. Strikingly, phage-resistant MRSA that survived ΦStaph1N infection displayed a strong reduction in resistance against OXA, CEF, and AMX + CA, with fold reductions in MIC between 10 and 1000-fold (*Figure 1D*); no change in MIC was observed with VANC or with AMX alone. These results show at a phenotypic level that ΦStaph1N-resistant MRSA loses resistance towards most β-lactams.

We next asked whether this loss of β-lactam resistance depended on the MOI of ΦStaph1N. We infected the three MRSA strains with ΦStaph1N at MOIs ranging from $10^{-2}$ to $10^{-5}$, isolated the surviving MRSA cells, and tested their MIC against oxacillin (*Table 1*). For MRSA252, we still observed a~3-order of magnitude fold reduction of MIC at an MOI of $10^{-5}$. With MW2, the reduction of MIC was

**Table 1.** Minimum inhibitory concentrations (µg/mL) against oxacillin of MRSA strains treated with different MOIs of phage.

**MRSA252**

| | ΦStaph1N | | | Evo2 | | |
|---|---|---|---|---|---|---|
| MOI | Rep 1 | Rep 2 | Rep 3 | Rep 1 | Rep 2 | Rep 3 |
| $10^{-2}$ | 0.25 | 0.125 | 0.38 | NG | 2 | 0.5 |
| $10^{-3}$ | NG | 0.94 | 0.19 | 1 | 0.75 | 1 |
| $10^{-4}$ | 0.5 | 0.25 | 0.19 | 0.75 | 1 | 0.5 |
| $10^{-5}$ | 0.25 | 0.38 | NG | 0.38 | NG | NG |
| Mock | >256 | >256 | >256 | >256 | >256 | >256 |

**MW2**

| | ΦStaph1N | | | Evo2 | | |
|---|---|---|---|---|---|---|
| MOI | Rep 1 | Rep 2 | Rep 3 | Rep 1 | Rep 2 | Rep 3 |
| $10^{-2}$ | 3 | 24 | 24 | 4 | NG | NG |
| $10^{-3}$ | 32 | 24 | 48 | 4 | NG | NG |
| $10^{-4}$ | 48 | 96 | 32 | 3 | NG | NG |
| $10^{-5}$ | 96 | 64 | 24 | 2 | NG | NG |
| Mock | 96 | 48 | 32 | 96 | 48 | 32 |

**LAC**

| | ΦStaph1N | | | Evo2 | | |
|---|---|---|---|---|---|---|
| MOI | Rep 1 | Rep 2 | Rep 3 | Rep 1 | Rep 2 | Rep 3 |
| $10^{-2}$ | NG | NG | 2 | 0.064 | NG | NG |
| $10^{-3}$ | NG | 3 | 1.5 | 0.032 | NG | NG |
| $10^{-4}$ | 32 | 1.5 | 1 | NG | NG | NG |
| $10^{-5}$ | 32 | 16 | 0.38 | NG | NG | NG |
| Mock | 32 | 48 | 48 | 32 | 48 | 48 |

NG: no growth detected.

markedly decreased with lower phage levels, showing no significant loss at MOIs of $10^{-3}$ or lower. For LAC, two replicates displayed a reduction of MIC by an order of magnitude at an MOI of $10^{-4}$, while the third replicate did not display any change. These results show that for MRSA252, ΦStaph1N MOIs as low as $10^{-5}$ can still drive the loss of resistance, while for MW2 and LAC, higher MOIs of phage are needed to ensure the same outcome of reduced β-lactam resistance.

## Discovery of a mutant ΦStaph1N with enhanced activity against MRSA

For ΦStaph1N, we noticed that while the phage could plaque on all three MRSA strains, its plaque-forming efficiency was reduced on the MW2 and LAC strains (*Figure 1B*; *Figure 1—figure supplement 1*). ΦStaph1N plaques on MW2 and LAC bacterial lawns were hazy, and the overall efficiency of plaquing was approximately two orders of magnitude less than that on MRSA252. Unexpectedly, we consistently observed smaller, clear plaques arising in the larger, hazy plaques of LAC (*Figure 2—figure supplement 1*); notably, this did not appear in MW2. We hypothesized that these clear plaques were caused by a mutant form of ΦStaph1N that evolved higher lytic activity. We isolated phage clones from these single plaques and tested their activity against MRSA. This mutant phage, which we called Evo2, plaques on LAC and MW2 strains with higher efficiency, displaying comparable plaquing to MRSA252 (*Figure 2A*). In growth experiments, we further observed that Evo2 lyses MRSA cultures at lower MOIs compared to ΦStaph1N (*Figure 2—figure supplement 2*). Evo2 exhibits lytic activity

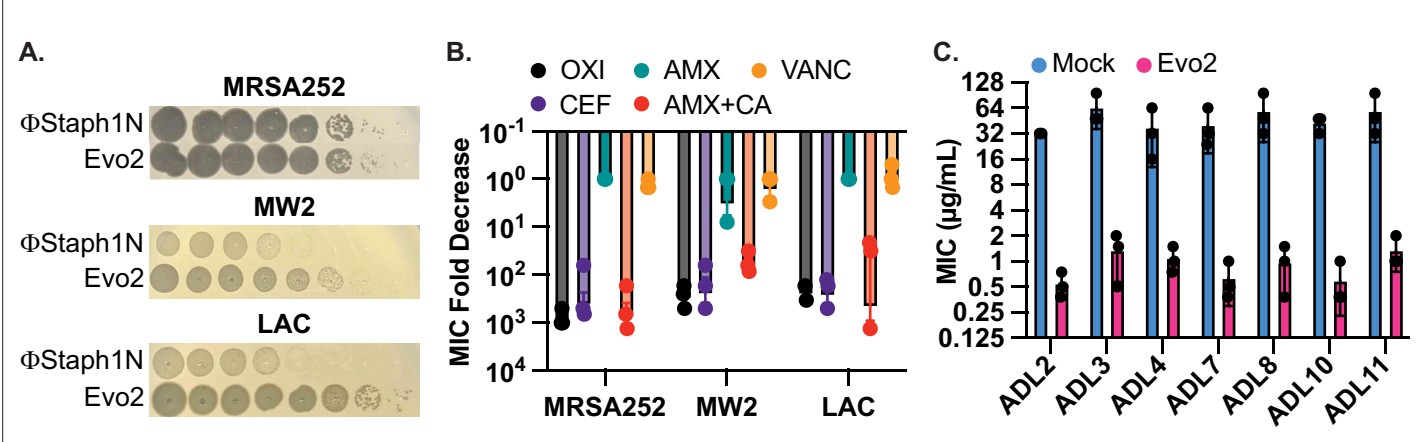

**Figure 2.** Evo2 is a variant of ΦStaph1N with higher activity against MRSA. (**A**) Evo2 shows comparable infectivity towards MRSA252 but improved infectivity towards MW2 and LAC, relative to ΦStaph1N. The same plaquing data is also shown in *Figure 2—figure supplement 1*. (**B**) Similar to ΦStaph1N, Evo2 infection reduces β-lactam resistance in MRSA. (**C**) Evo2 infection reduces the MIC against oxacillin in clinical isolates of USA300 (ADLs). All error bars represent the Standard Error of the Mean (SEM) of three independent replicates.

The online version of this article includes the following source data and figure supplement(s) for figure 2:

**Source data 1.** Uncropped plate images for *Figure 2A*.

**Source data 2.** Source data for the bar graphs in *Figure 2B and C*.

**Figure supplement 1.** Isolation and sequencing analysis of Evo2.

**Figure supplement 1—source data 1.** Uncropped plate images for *Figure 2—figure supplement 1A*.

**Figure supplement 2.** Growth curves of MRSA strains under varying levels of Evo2 infection.

**Figure supplement 3.** Phage ΦNM1γ6 infection LAC does not drive the loss of β-lactam resistance.

**Figure supplement 3—source data 1.** Uncropped plate images for *Figure 2—figure supplement 3A and B*.

**Figure supplement 4.** Phage SATA8505 infection drives loss of oxacillin resistance.

**Figure supplement 4—source data 1.** Uncropped plate images for *Figure 2—figure supplement 4A and B*.

against MW2 and LAC even at an MOI of $10^{-4}$, a concentration at which ΦStaph1N does not show any detectable activity against the two strains.

We sequenced the genome of Evo2 to determine the genetic mechanism driving this enhanced activity. We observed a single point mutation in ORF141 that induces a premature stop codon (*Figure 2—figure supplement 1*). Sequence analysis with HHpred predicts ORF141 to be a putative DNA binding protein with an HTH motif (PDB: 2LVS, E-value: 2.5e-9). We speculate that this protein is a transcriptional regulator that, when inactivated by a nonsense mutation, increases ΦStaph1N infectivity. Future studies will center on determining the mechanism of this mutation and why Evo2 only evolved in the LAC strain.

Given Evo2's enhanced activity against MRSA, we asked how predation by Evo2 affected β-lactam resistance. We infected MRSA252, MW2, and LAC with Evo2 at an MOI of 0.1 and measured the MICs against β-lactams after 48 hours of passaging. Similar to ΦStaph1N, infection by Evo2 reduced the MICs of the three MRSA strains against OXA, CEF, and AMX + CA, while MICs against AMX alone and VAN did not change significantly (*Figure 2B*). Expanding on the different classes of antibiotics, we tested whether Evo2 predation could impact the susceptibility to the transcription inhibitor rifampicin; the translation inhibitors erythromycin and mupirocin; and the cell envelope disruptors teicoplanin, fosfomycin, and daptomycin (*Table 2*). We found that the MICs of these antibiotics did not change significantly, with a few exceptions: in some cases, Evo2-resistant LAC became sensitized to fosfomycin and daptomycin; furthermore, one replicate of Evo2-resistant MRSA252 evolved sensitivity to teicoplanin. However, overall, the MIC reduction in these cases was not as dramatic as the MIC reduction seen against β-lactams.

Finally, we examined how different MOIs of Evo2 impacted β-lactam resistance in MRSA (*Table 1*). We infected the three MRSA strains with Evo2 at varying MOIs from $10^{-2}$ to $10^{-5}$ and measured the MIC against oxacillin of the evolved MRSA. Notably, across all three strains, we found that multiple

**Table 2.** Minimal inhibitory concentrations (µg/mL) of mock- or Evo2-treated MRSA strains against different antibiotics.

| Strain | Antibiotic | Mock | | | Evo2 | | |
|---|---|---|---|---|---|---|---|
| | | Rep 1 | Rep 2 | Rep 3 | Rep 1 | Rep 2 | Rep 3 |
| | Oxacillin | >256 | >256 | >256 | 0.38 | 0.75 | 0.5 |
| | Rifampicin | 0.047 | 0.032 | 0.012 | 0.023 | 0.047 | 0.023 |
| | Mupirocin | 0.75 | 0.5 | 1 | 0.5 | 0.5 | 0.38 |
| | Erythromycin | >256 | >256 | >256 | >256 | >256 | >256 |
| | Teicoplanin | 6 | 6 | 4 | 4 | 4 | 0.75 |
| | Fosfomycin | 12 | 8 | 8 | 8 | 8 | 6 |
| MRSA252 | Daptomycin | 2 | 3 | 2 | 2 | 2 | 2 |
| | Oxacillin | 48 | 32 | 32 | 0.75 | 1 | 0.75 |
| | Rifampicin | 0.032 | 0.047 | 0.064 | 0.023 | 0.023 | 0.032 |
| | Mupirocin | 0.5 | 0.25 | 0.5 | 0.38 | 0.38 | 0.25 |
| | Erythromycin | 1 | 0.75 | 0.75 | 0.25 | 0.5 | 0.25 |
| | Teicoplanin | 2 | 1.5 | 1.5 | 1.5 | 1 | 1 |
| | Fosfomycin | 1 | 1.5 | 1.5 | 0.5 | 1 | 1 |
| MW2 | Daptomycin | 1.5 | 2 | 1.5 | 0.75 | 0.5 | 3 |
| | Oxacillin | 48 | 24 | 64 | 0.19 | 0.064 | 0.047 |
| | Rifampicin | 0.047 | 0.047 | 0.047 | 0.032 | 0.032 | 0.047 |
| | Mupirocin | 0.5 | 0.75 | 0.75 | 0.5 | 0.5 | 0.5 |
| | Erythromycin | 3 | 3 | 3 | 1.5 | 1 | 2 |
| | Teicoplanin | 1 | 0.5 | 1 | 0.5 | 0.5 | 0.75 |
| | Fosfomycin | 6 | 6 | 12 | 1.5 | 12 | 1 |
| LAC | Daptomycin | 0.5 | 3 | 1 | 0.064 | 2 | 0.75 |

Rep = biological replicate.

replicate cultures across different MOIs were unable to recover growth following Evo2 infection. However, cultures of MRSA252, MW2 and LAC that did regrow displayed a loss of oxacillin resistance, between 10- and 1000-fold. Thus, overall Evo2 displayed a higher infectivity against MRSA and a greater potency in reducing β-lactam resistance.

## Evo2 is broadly active against recent clinical isolates of *S. aureus* USA300

MRSA252, MW2, and LAC were isolated in 1997, 1998, and in the early 2000 s, respectively. We therefore tested if Evo2 can infect more recent *S. aureus* clinical isolates. We compared the plaquing efficiency of Evo2 and ΦStaph1N against 30 USA300 strains that were isolated between 2008 and 2011 at St. Louis Children's Hospital (*Land et al., 2015*). We observe dramatic variation in the plaquing efficiency of ΦStaph1N and the 30 strains, while Evo2 exhibited a higher plaquing efficiency in the majority of the 30 strains (*Table 3*). We then tested how infection of Evo2 impacted OXA resistance in 12 (ADL1-12) of these clinical isolates. We infected the strains with Evo2 at an MOI of 0.1, and if a phage-resistant population was recovered, we measured the OXA MIC after 48 hr of passaging. Interestingly, after 15 independent challenges with Evo2, we were not able to recover phage-resistant populations from ADL1, 5, 6, and 12. This suggests that Evo2 resistance acquisition is a rare event in these strains. In the rest of the strains, we observed that similarly to MRSA252, MW2, and LAC, the OXA MIC was reduced between 10- and 100-fold after Evo2 infection (*Figure 2C*). Overall, these results highlight the broader host range and activity of Evo2.

**Table 3.** Efficiencies of plaquing (EOPs)* of ΦStaph1N, Evo2, and ΦNM1$\gamma$6 on clinical isolates of USA300 (ADL1-30).

| Strain | ΦStaph1N | Evo2 | ΦNM1γ6 |
|---|---|---|---|
| RN4220 | 1.0E+00 | 1.0E+00 | 1.0E+00 |
| ADL1 | 1.7E-02 | 2.7E+00 | 6.7E-01 |
| ADL2 | 2.0E-01 | 3.3E+00 | 1.0E+00 |
| ADL3 | 6.0E-02 | 2.0E+00 | 1.7E-02 |
| ADL4 | 1.3E-03 | 1.0E+00 | 1.7E-01 |
| ADL5 | 1.2E-02 | 2.7E+00 | 1.0E+00 |
| ADL6 | 1.5E-02 | 1.7E+00 | 1.0E+00 |
| ADL7 | 6.0E-01 | 1.7E+00 | 6.7E-01 |
| ADL8 | 9.0E-02 | 2.0E+00 | 3.3E-01 |
| ADL9 | 6.0E-03 | 1.3E+00 | 2.0E-04 |
| ADL10 | 5.0E-01 | 1.3E+00 | 1.0E-03 |
| ADL11 | 1.0E+00 | 1.0E+00 | 3.3E-01 |
| ADL12 | 9.0E-04 | 2.0E+00 | 6.7E-07 |
| ADL13 | 7.0E-02 | 2.7E+00 | 2.3E-01 |
| ADL14 | 2.7E-01 | 3.3E+00 | 1.0E-05 |
| ADL15 | 9.0E-02 | 3.0E+00 | 2.0E-02 |
| ADL16 | 2.4E-01 | 5.6E+00 | 7.8E-02 |
| ADL17 | 3.3E+00 | 1.0E+01 | 2.4E+00 |
| ADL18 | 6.7E+00 | 1.4E+01 | 1.6E-02 |
| ADL19 | 7.1E-01 | 1.6E+00 | 2.3E+00 |
| ADL20 | 6.2E+00 | 7.8E+01 | 2.4E+00 |
| ADL21 | 5.2E+00 | 5.6E+00 | 3.3E-06 |
| ADL22 | 9.5E-01 | 2.0E+00 | 5.2E-01 |
| ADL23 | 1.5E+00 | 3.3E+01 | 1.2E-01 |
| ADL24 | 7.1E-01 | 5.6E+00 | 3.3E-06 |
| ADL25 | 7.1E-01 | 1.8E+00 | 2.3E-01 |
| ADL26 | 2.4E-02 | 3.9E+00 | 2.2E-03 |
| ADL27 | 1.3E+00 | 7.8E+00 | 1.7E+00 |
| ADL28 | 1.9E+00 | 8.9E+00 | 8.9E-02 |
| ADL29 | 1.9E+00 | 1.0E+01 | 1.3E-04 |
| ADL30 | 6.7E-01 | 1.4E+00 | 1.7E-05 |

*Phage EOPs on the clinical isolates are standardized to their respective EOP on the laboratory strain *S. aureus* RN4220.

## Effect on β-lactam MIC by other staphylococcal phages

We asked whether two additional phages from our collection could elicit MIC reduction against oxacillin in MRSA. The MRSA LAC strain is sensitive to infection by ΦNM1γ6, a lytic version of the temperate phage ΦNM1 of the *Dubowvirus* genus, derived from the *S. aureus* Newman strain (*Mo et al., 2021*; *Goldberg et al., 2014*; *Baba et al., 2008*). In LAC, ΦNM1γ6 displays plaquing comparable to that of ΦStaph1N, while also showing activity against some of the clinical USA300 isolates (*Figure 1—figure supplement 1*; *Table 3*). Therefore, we infected LAC with ΦNM1γ6 at an MOI of 0.1 and measured the MIC against β-lactams of the surviving cells. While the recovered LAC cultures

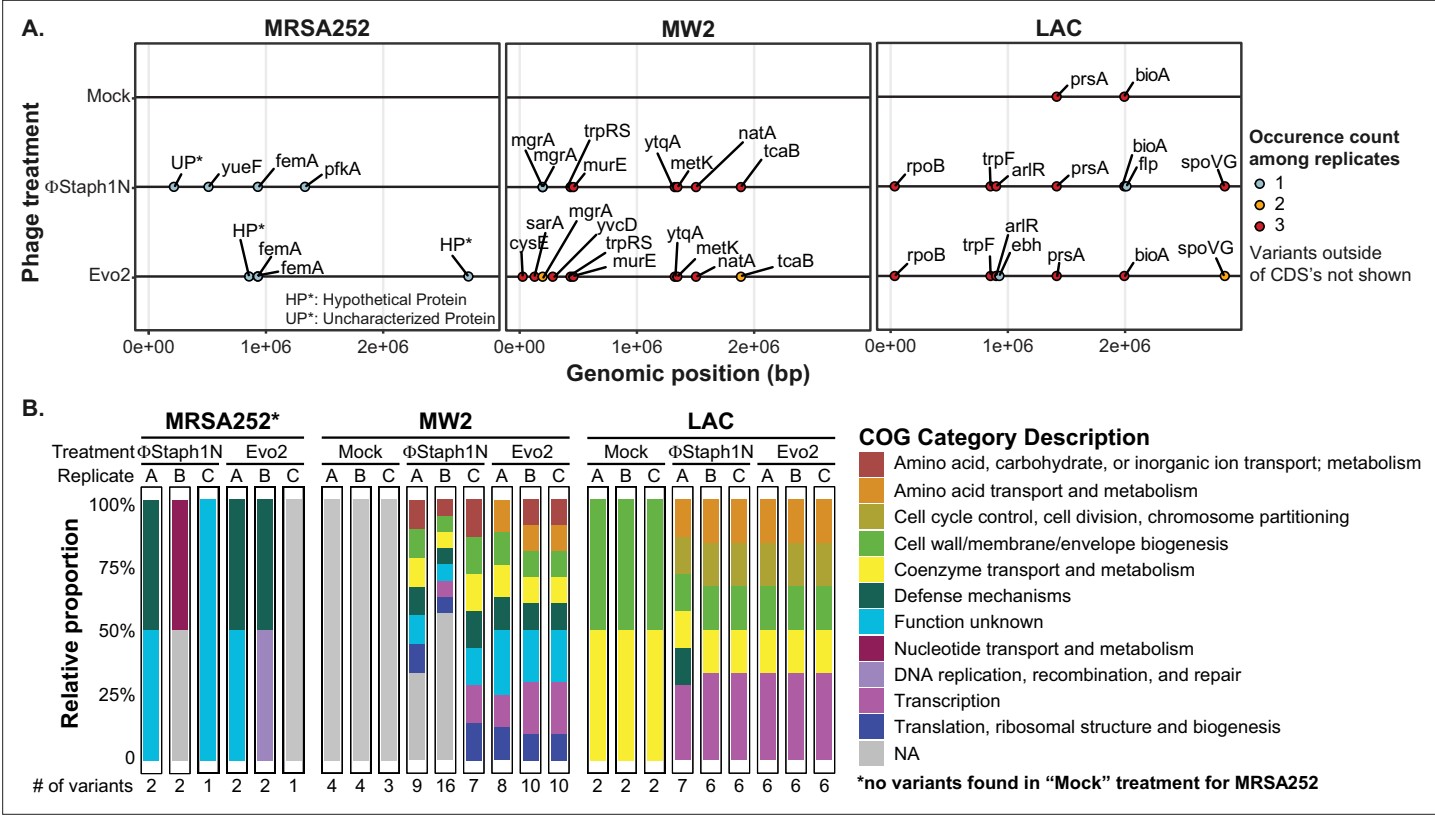

**Figure 3.** Phage infection of MRSA strains produces distinct mutational profiles. (**A**) Coding sequences (CDS) with mutations from the three MRSA strains following phage treatment or mock treatment. For each strain, three isolates were sequenced and their mutations identified. Mutations are color-coded based on the number of occurrences among the three replicates. Information on all detected genetic variants is listed in **Supplementary file 1**. (**B**) Categories of genes with mutations that arose in each MRSA strain and treatment condition.

The online version of this article includes the following source data and figure supplement(s) for figure 3:

**Figure supplement 1.** Types of polymorphisms in MRSA strains following infection by phage or a mock treatment.

**Figure supplement 2.** Plaquing efficiency of Evo2 and ΦStaph1N on MW2 and LAC strains with knockouts in mgrA, arl, and sarA.

**Figure supplement 2—source data 1.** Uncropped plaquing images of **Figure 3—figure supplement 2**.

exhibited resistance against ΦNM1γ6, they did not show a reduction in MIC against the panel of β-lactams (**Figure 2—figure supplement 3**), suggesting that phage resistance caused by ΦNM1γ6 is uncoupled from β-lactam resistance. We also isolated from the environment a second phage of the *Kayvirus* genus, called SATA8505 (**Pincus et al., 2015**), and tested its activity against MRSA. SATA8505 is active against MRSA252, MW2, and LAC (**Figure 2—figure supplement 4**), and infection of the three strains caused a rise of phage resistance in MRSA252, MW2, and LAC. Similar to ΦStaph1N and Evo2, cells resistant to SATA8505 showed a strong loss of oxacillin resistance (**Figure 2—figure supplement 4**). Although a more comprehensive study is needed, our results with these two phages suggest that the ability to reduce β-lactam resistance is not a universal feature across all staphylococcal phages and call for a more comprehensive analysis of staphylococcal phages and their ability to elicit β-lactam trade-offs.

## Genomic mutations in MRSA strains following phage infection

Following our phenotypic analyses, we examined the genomes of the phage-resistant MRSA. We first sequenced the genomes of three clonal isolates (A-C) from each MRSA strain that underwent ΦStaph1N, Evo2, or a mock infection. We observed that each MRSA strain evolved distinct mutation profiles (**Figure 3A**). Irrespective of the strain and phage treatment, most mutations were predicted to be substitutions, followed by truncations (**Figure 3—figure supplement 1**). Cluster of Orthologous Genes (COG) variants associated with transcription, cell wall/membrane/envelope biogenesis,

**Table 4.** Mutated genes in MRSA following infection with phages ΦStaph1N or Evo2.

| Gene | Description | Strain | Phage infection/treatment | Reference |
|---|---|---|---|---|
| sarA | Transcriptional regulator of antibiotic resistance and virulence | MW2 | Evo2 | *Li et al., 2016*; *Zielinska et al., 2012* |
| mgrA | Transcriptional regulator of antibiotic resistance and virulence | MW2 | ΦStaph1N, Evo2 | *Crosby et al., 2016*; *Kwiecinski et al., 2021* |
| rpoB | Beta subunit of RNA polymerase Transcriptional regulator of antibiotic resistance | LAC | ΦStaph1N, Evo2 | *Panchal et al., 2020* |
| arlR | Transcriptional regulator of antibiotic resistance and virulence | LAC | ΦStaph1N, Evo2 | *Kwiecinski et al., 2021*; *Bai et al., 2019*; *Walker et al., 2013* |
| spoVG | Transcriptional regulator of antibiotic resistance and virulence | LAC | ΦStaph1N, Evo2 | *Schulthess et al., 2011*; *Liu et al., 2016* |
| cysE | Cysteine and methionine synthesis, serine O-acetyltransferase | MW2 | Evo2 | *Chen et al., 2019* |
| metK | Cysteine and methionine synthesis, S-adenosylmethionine (SAM) synthetase | MW2 | ΦStaph1N, Evo2 | *Markham et al., 1984* |
| trpF | Phenylalanine, tyrosine and tryptophan synthesis, phosphoribosylanthranilate isomerase | LAC | ΦStaph1N, Evo2 | *Proctor and Kloos, 1973* |
| femA | Peptidoglycan synthesis, pentaglycine synthesis | MRSA252 | ΦStaph1N, Evo2 | *Maidhof et al., 1991*; *Srisuknimit et al., 2017* |
| murE | Peptidoglycan synthesis, UDP-MurNAc tripeptide synthesis | MW2 | ΦStaph1N, Evo2 | *Gardete et al., 2004* |
| trpS | Aminoacyl-tRNA synthesis, tryptophanyl-tRNA synthesis | MW2 | ΦStaph1N, Evo2 | *Xu et al., 1989* |
| ytqA | tRNA modifications, $mnm^5s^2U$ synthesis | MW2 | ΦStaph1N, Evo2 | *Jaroch et al., 2024* |
| yvcD | Unknown | MW2 | Evo2 | |
| natA | ABC transporter | MW2 | ΦStaph1N, Evo2 | *Kobayashi et al., 2001* |
| tcaB | Predicted multidrug efflux pump | MW2 | ΦStaph1N, Evo2 | *Maki et al., 2004* |
| fmhC | Fem-like factors | LAC | ΦNM1γ6 | *Willing et al., 2020* |
| rsaC ncRNA | modulates oxidative stress response and metal immunity | MW2 | ΦStaph1N+oxacillin | *Lalaouna et al., 2019* |
| nrdF | class 1b ribonucleoside-diphosphate reductase subunit beta; beta subunit contains a metal-based cofactor; involved in DNA synthesis | MW2 | ΦStaph1N+oxacillin | *Masalha et al., 2001* |
| fstAT ncRNA | Unknown | MW2 | ΦStaph1N+oxacillin | |
| rpoC | DNA-directed RNA polymerase subunit beta' | MW2 | ΦStaph1N+oxacillin | |
| tRNA | Transfer RNA | MW2 | ΦStaph1N+oxacillin | |

coenzyme transport and metabolism, and defense mechanisms were the most commonly found categories. Mutations in annotated genes that appeared at least twice across the clonal replicates are summarized in *Table 4*. Information on all detected genetic variants is listed in *Supplementary file 1*.

One plausible hypothesis explaining the loss of β-lactam resistance is that phage infection selected for a defective *SCCmec* or *blaZ*. However, we did not observe any mutations in the two loci. Instead, all MRSA strains exhibited mutations in ancillary genes implicated in the loss of β-lactam resistance. In MRSA252, both ΦStaph1N and Evo2 infection were selected for frameshift or nonsense mutations in the *femA* gene that would inactivate the protein product. As discussed above, *femA* is required for the synthesis of the pentaglycine branch on *S. aureus* Lipid II, the peptidoglycan precursor (*Table 4*). Deletions of *femA* have been shown to increase susceptibility to β-lactams even when PBP2a (encoded by *mecA*) is expressed, thus providing a genetic mechanism for how some MRSA252 cells lose β-lactam resistance after phage selection (*Maidhof et al., 1991*; *Srisuknimit et al., 2017*). At the same time, we found the presence of other uncharacterized mutations in phage-resistant MRSA252. For example,

clone A of ΦStaph1N-treated cells carried two mutations: a frameshift in *femA* and a substitution mutation in an uncharacterized protein; meanwhile, clone B displayed a substitution mutation in *pfkA*, a predicted ATP-dependent 6-phosphofructokinase and mutation in an intergenic region; clone C showed a substitution mutation in a putative transport protein, called *yueF* (*Figure 3B*, *Supplementary file 1*). The role of these mutations in mediating phage resistance or β-lactam sensitivity, if any, remains unknown.

In MW2, we found mutations in two transcriptional regulators, *mgrA* and *sarA* (*Figure 3A*, *Table 4*). Both *mgrA* and *sarA* belong to the family of MarR (multiple antibiotic resistance regulator)/SarA (staphylococcal accessory regulator A) proteins, which regulate drug resistance and virulence in *S. aureus* (*Li et al., 2016*; *Zielinska et al., 2012*; *Crosby et al., 2016*). In ΦStaph1N-treated MW2, only *mgrA* was mutated, while in Evo2-treated MW2, clones also showed nonsense mutations in *sarA*. We also found mutations in *metK* and *ytqA,* which are both predicted to be associated with S-adenosylmethione (SAM): *metK* synthesizes SAM, while *ytqA* belongs to the radical SAM enzyme family and is predicted to be involved in tRNA modification (*Markham et al., 1984*; *Jaroch et al., 2024*). Phage-treated MW2 also displayed mutations in other genes, including *tcaB and murE*. Notably, each clonal replicate had multiple mutations in the genome, while by contrast, untreated MW2 cells only displayed a deletion in an intergenic region that is not present in any of the phage-treated samples. These findings suggest that MW2 could be amassing multiple mutations during the course of phage infection.

For LAC, we observed a third, distinct mutational pattern (*Figure 3A*). Of note, we found nonsense mutations in *arlR,* which is part of the *arlRS* two-component signaling system (*Table 4*). The activity of *arlRS* has been implicated in *S. aureus* virulence, pathogenicity, and oxacillin resistance (*Bai et al., 2019*; *Walker et al., 2013*). Further, we observed substitution mutations in *spoVG*, which is a transcription factor regulating the expression of genes involved in a variety of functions, including cell wall metabolism (*Table 4*; *Schulthess et al., 2011*). Indeed, *spoVG* activates the expression of *femA* (*Liu et al., 2016*). Studies have shown that *spoVG* modulates β-lactam antibiotic resistance by modulating cell wall synthesis (*Liu et al., 2016*). Similar to the other two MRSA strains, phage-treated clones of LAC showed multiple mutations in their genomes. We observed mutations in *prsA* and *bioA* that appeared in both the mock and phage treatment conditions, suggesting that these mutations do not arise due to phage selection.

Altogether, our results show that phage-infected MRSA strains acquire distinct mutational profiles. These mutations likely work in concert to promote phage resistance and β-lactam sensitivity, making it challenging to determine the mechanistic contributions of individual mutations. For example, we observed that the genes *mgrA*, *sarR*, and *arlR* evolved nonsense mutations, which would result in truncated, potentially non-functional protein products. We therefore tested if single knockout mutants of these genes alone are sufficient to confer resistance to ΦStaph1N and Evo2 (*Figure 3—figure supplement 2*). In MW2, the *mgrA* knockout resulted in a modest reduction in plaquing of Evo2 and ΦStaph1N. However, none of the remaining mutants in either the MW2 or LAC background conferred resistance. Prior experimental studies have also shown that phage resistance in *S. aureus* can arise from the disruption of single genes directly involved in the synthesis and modification of WTA, such as *tagO* (*Jurado et al., 2022*). Some MRSA strains also alter cell wall glycosylation through dedicated genes encoded on prophages (*Gerlach et al., 2018*). However, we did not see any mutations in genes directly involved in WTA synthesis. Our results thus highlight how MRSA can take on unique mutational pathways under phage selection.

Finally, we examined the mutational profile in ΦNM1γ6−resistant LAC populations. Because infection with ΦNM1γ6 did not result in a decrease in OXA resistance, we hypothesized that mutations that arose in LAC following ΦNM1γ6 would be distinct from those following Evo2 infection. We found that the two genes were mutated across three clonal isolates from different resistant populations: *bioA* and *fmhC*. As seen previously, mutations in *bioA* appeared in the mock treatment, suggesting that the mutations arose independently of phage selection. On the other hand, LAC showed a missense mutation in *fmhC* (H21D, *Table 4*, *Supplementary file 1*). *FmhC* and its homologue *fmhA* pair with *femA* and *femB* to incorporate Gly-Ser dipeptides into peptidoglycan cross-bridges (*Willing et al., 2020*). However, the mechanism of the H21D mutation is unknown, and to our knowledge, mutations in *fmhC* have not been associated with phage resistance in *S. aureus* before.

## Phage-treated MRSA strains display broad changes to their transcriptomes

Our genomic analysis revealed that phage-treated MRSA evolved mutations in a variety of transcriptional regulators, some of which are known to affect MRSA virulence. We therefore hypothesized that the mutations in these regulators would fundamentally alter the transcriptional profile of the treated MRSA. To test this, we performed bulk RNA-seq experiments on MW2 and LAC strains that were treated with the phage Evo2 and compared their transcription profiles to those of untreated strains (*Figure 4*, *Supplementary file 2*). We observed significant changes in gene expression in both MW2 and LAC. Notably, mirroring the trend seen in the mutational data, we did not observe significant changes in the expression of genes in the *SCCmec* cassette or *blaZ* present in both MW2 and LAC.

We first compared our expression data against transcriptomic studies from previous studies. For example, Horswill and colleagues have shown that deletions of *arlRS* and *mgrA* de-repress the extracellular matrix binding protein *ebh*, resulting in significantly higher expression levels of the gene. In addition, these deletions also increased the expression of urease genes involved in the urea TCA cycle (*Crosby et al., 2016*; *Walker et al., 2013*). In our experiments, both MW2 and LAC strains evolved nonsense mutations in *mgrA* and *arlR,* respectively. We therefore hypothesized that these mutations would mimic the effects of gene deletions and likewise yield elevated transcript levels of *ebh* and urease genes. Aligning with our hypothesis, differential expression data from both MW2 and LAC displayed $\log_2$ fold changes of >7 for *ebh* and >3 for *ureABCDEFG* genes (*Supplementary file 2*).

Additionally, both MW2 and LAC strongly upregulated several genes involved in cell wall maintenance. These include *lytN*, which is a murein hydrolase involved in the cross-wall compartment of *S. aureus*, and *fmhC*, which, as described previously, incorporates Gly-Ser dipeptides into pentaglycine cross-bridges in the *S. aureus* peptidoglycan cell wall. Overexpression of *lytN* has been shown to damage the cell wall, which in turn is alleviated by overexpression of *fmhC* (*Willing et al., 2020*). Interestingly, *fmhC* overexpression is linked to increased β-lactam sensitivity and thus may contribute to the loss of β-lactam resistance phenotypes we observed in the MRSA strains.

MW2 and LAC also downregulated numerous genes, many of which are known virulence factors. Both strains reduced transcript levels of genes in the locus of the type VII secretion system (*ess* locus), staphyloxanthin biosynthesis (*crtM, crtN, crtP*), and quorum sensing (e.g. *agrA*; *Figure 4*, *Supplementary file 2*). Individually, these pathways have been shown to bolster the ability of *S. aureus* to establish infection and evade the host immune system (*Cao et al., 2016*; *Clauditz et al., 2006*; *Yarwood and Schlievert, 2003*). Our results suggest that infection by Evo2 can lead MRSA to reduce the expression of all of these pathways, which could reduce the virulence of *S. aureus.*

Finally, we noted that both MW2 and LAC showed transcriptional changes that appear to be strain specific. For example, MW2 saw a significant increase in the virulence factor *spa*, known to interfere with the host immune response and interface with other bacterial species. The presence of cell wall-bound Protein A has also been shown to decrease phage absorption, likely by masking WTA (*Moller et al., 2021*). Further in MW2, we found that *tarM*, which adds α1,4-GlcNAc to WTA, was strongly up-regulated ($\log_2$ ratio = 6.63). This is in line with previous findings showing that elevated *tarM* and α1,4-GlcNAc-WTA can lead to phage resistance in MRSA. The LAC showed an increase in the expression ($\log_2$ fold change >3.8) of the hemolytic cytotoxin genes *lukD/E,* which lyses host cells and targets neutrophils (*Reyes-Robles et al., 2013*). We do not know whether the increased expression of these genes results in a greater level of protein production and secretion, but these transcriptional changes could represent a potential 'trade-up' associated with phage resistance. Additional studies will be needed to fully assess the physiological and ecological effects of these upregulated genes in MRSA. Altogether, our RNA-seq results suggest that phage infection and resistance in MRSA cause significant transcriptional changes across a wide range of virulence, metabolic, and cell-wall-associated genes.

## Phage-treated MRSA strains display reduced virulence phenotypes

*S. aureus* is a highly virulent pathogen, relying on a vast array of toxins and immune evasion proteins to promote infection (*Cheung et al., 2021*). MW2 and LAC strains are models of MRSA virulence (*Crosby et al., 2016*; *Kwiecinski et al., 2021*). In light of our mutational and transcriptomic data, we hypothesized that phage-treated MRSA cells would display altered virulence phenotypes, in addition to reduced β-lactam resistance. We first tested the ability of MRSA strains that survived phage predation for their ability to form biofilms in a Crystal Violet assay. We found that Evo2 infection of

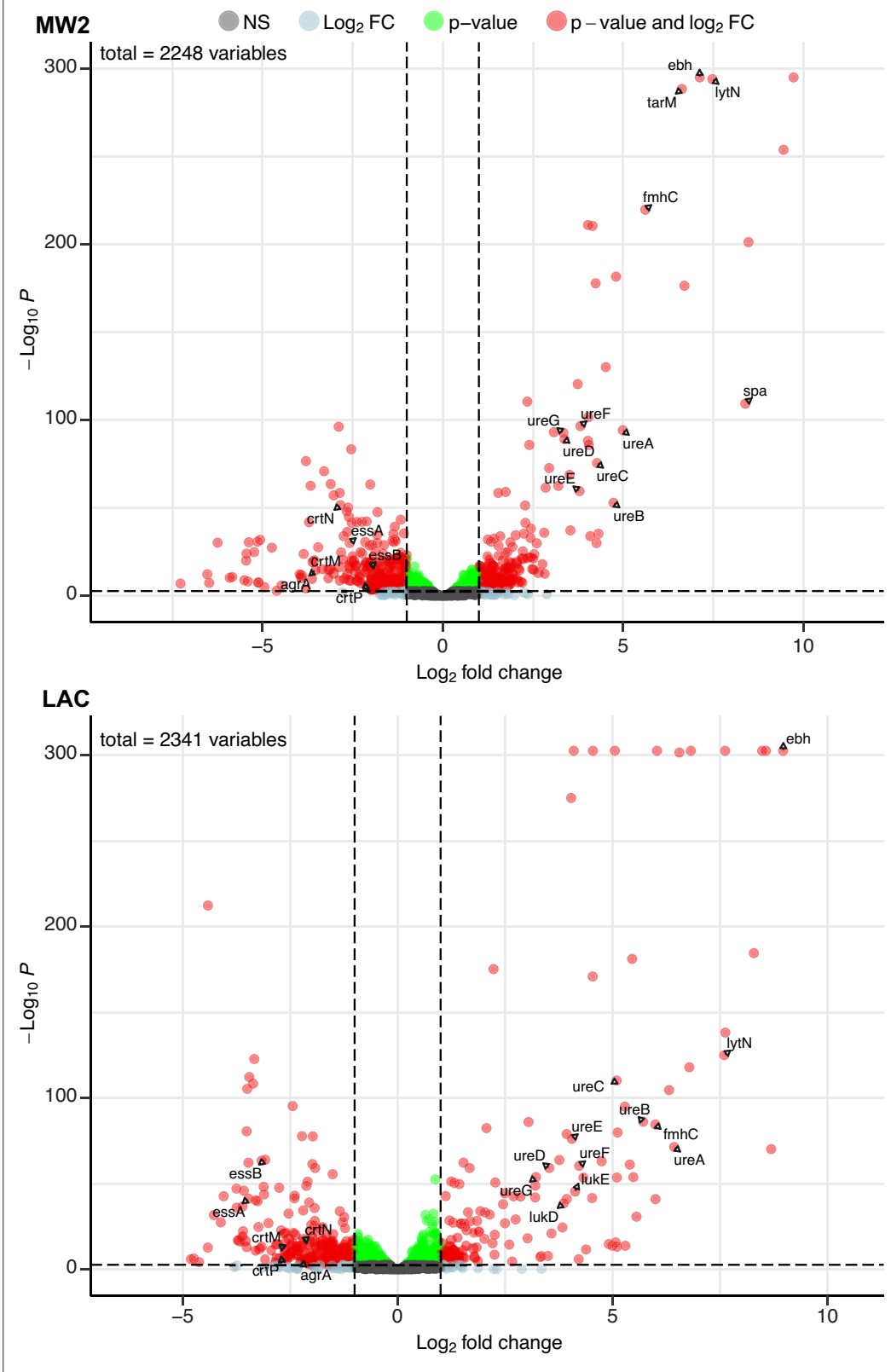

**Figure 4.** Phage infection changes the transcriptomic profile of MRSA. Differential expression analysis was performed on the transcriptomes of MW2 (top panel) and LAC (bottom panel). For both strains, Evo2-infected samples were compared to uninfected controls. Three biological replicates were analyzed for each condition. Horizontal dotted lines represent an adjusted p-value cut-off of 0.002, while vertical dotted lines represent a log$_2$

*Figure 4 continued on next page*

*Figure 4 continued*

fold change of –2 or 2 in expression. Transcripts with a log$_2$ fold change between –2 or 2 and a pvalue >0.002 are labeled as grey dots (Not significant, NS); transcripts that pass either the fold change or p-value cutoff (but not the other) are represented as blue and green dots, respectively; transcripts that pass both cutoffs are shown as red dots. Genes discussed in the main text are labeled. Data for all the transcripts with significant fold changes is shown in **Supplementary file 2**.

MRSA252 resulted in a significant reduction in Crystal Violet absorption compared to the parental strain. However, we show no significant difference in Crystal Violet absorption between parental, mock- and phage-treated MW2 and LAC strains (**Figure 5—figure supplement 1**).

Next, we tested whether phage infection could affect the hemolysis of rabbit blood cells. Hemolysis is mediated by the secretion of toxins, notably alpha toxin encoded by the gene *hla*, and plays an important role in MRSA infection (**Otto, 2014**). Expression of these toxins is regulated by virulence pathways that comprise numerous transcription factors, including *mgrA*, *arlR*, and *sarA*. Furthermore, in our RNA-seq results, we found that phage-resistant MRSA strains showed reduced expression of other cytotoxins. Parental MW2 and LAC colonies lysed rabbit blood cells on blood agar plates, producing distinct halos of clearance around the bacterial cells. For untreated LAC, the total area of hemolysis was on average 3-fold larger than that of untreated MW2 (~210 mm$^2$ vs ~70 mm$^2$,

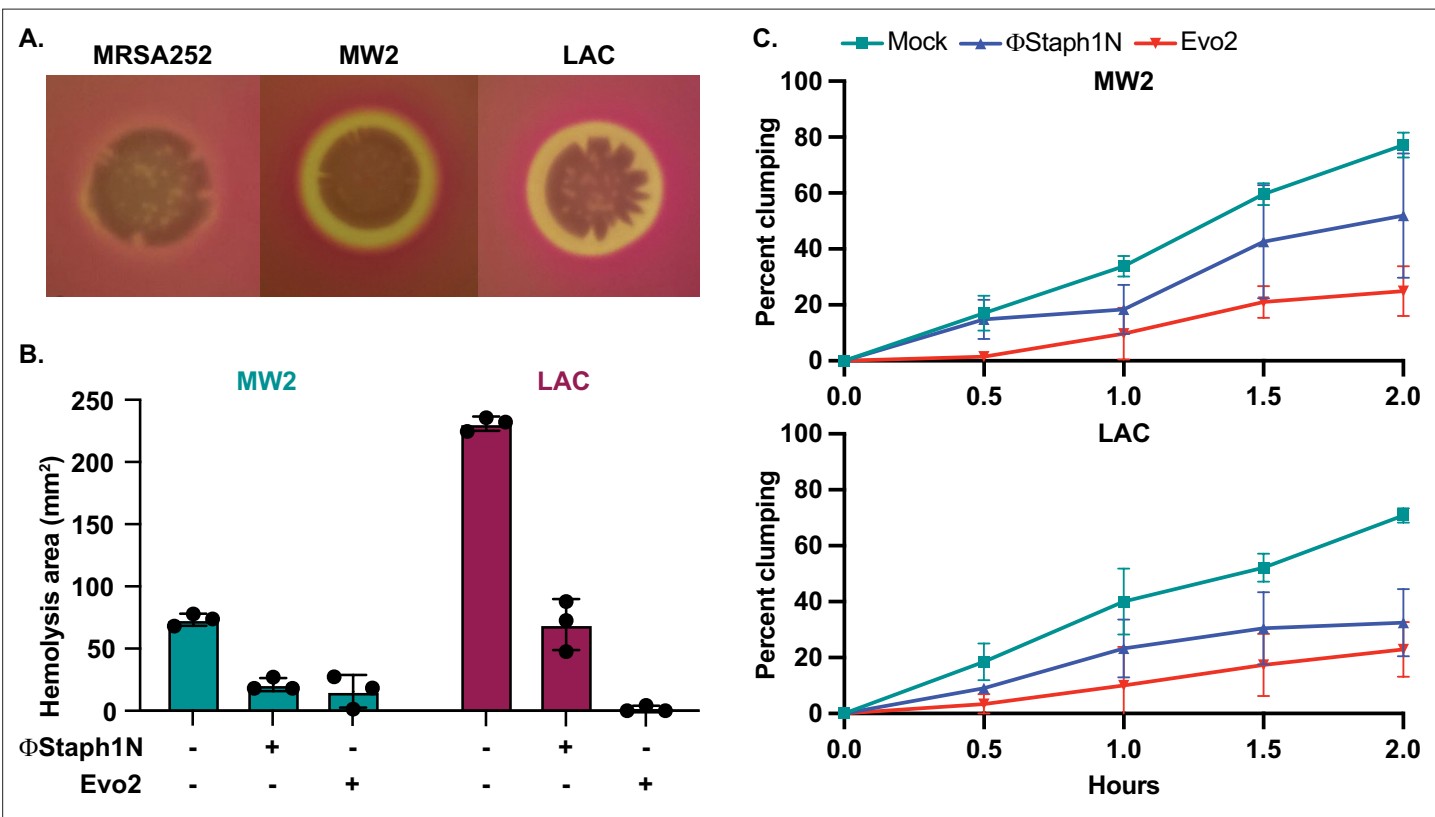

**Figure 5.** Phage treatment of MRSA results in attenuated virulence phenotypes. (**A**) MW2 and LAC strains display hemolytic activity on rabbit blood agar plates, while MRSA252 does not. (**B**) Phage-treated MW2 and LAC strains display reduced hemolysis compared to uninfected cells. (**C**) Surviving cultures of MW2 and LAC treated with either ΦStaph1N (blue) or Evo2 (red) show reduced clumping rates compared to mock untreated cells (teal). Each condition was tested in three independent replicates and error bars represent the Standard Deviation (SD).

The online version of this article includes the following source data and figure supplement(s) for figure 5:

**Source data 1.** Uncropped plates images for *Figure 5A*.

**Source data 2.** Source data for the bar graphs in *Figure 5B*.

**Figure supplement 1.** Effect of phage infection on biofilm formation in MRSA strains.

**Figure supplement 1—source data 1.** Source data for the bar graphs in *Figure 5—figure supplement 1*.

respectively); with MRSA252, by contrast, lysis was not detected (*Figure 5A*). Following treatment with ΦStaph1N, we observed that MW2 and LAC displayed a reduced area of hemolysis by four- to fivefold. With Evo2-treated cells, we found that in MW2, the fold reduction was comparable to that of ΦStaph1N-treated cells. However, for Evo2-treated LAC, loss of hemolysis was even more pronounced, with two of the replicates showing no detectable hemolysis. We note that neither MW2 nor LAC showed a reduction in transcript expression of *hla*. We posit that the loss of hemolysis could be driven by an inability of phage-evolved MRSA to secrete the toxin.

We next tested how phage infection affected cell agglutination (or clumping) in MRSA. *S. aureus* binds to fibrinogen, forming protective aggregates of bacterial cells. Clumping is thought to have several functions in the context of staphylococcal infections, facilitating adhesion to the host tissue. Clumps are also likely to be more resistant to clearance by the immune system, partly because they may be too large to be phagocytosed by neutrophils (*Kwiecinski et al., 2021*). In our transcriptional data, we noted that several cell surface proteins known to reduce cell clumping, such as *ebh*, were over-expressed in phage-resistant MRSA. Horswill and colleagues found that de-repression of *ebh* reduces clumping. Indeed, phage-treated MW2 and LAC displayed less clumping than the mock-treated or parental strain. For MW2, ΦStaph1N infection resulted in a modest reduction, while Evo2 infection resulted in a reduction of approximately 3-fold (*Figure 5B*). In LAC, we found that both ΦStaph1N and Evo2 treatment resulted in comparable reductions of clumping in surviving cells. Overall, these phenotypic results align with our genetic and transcriptomic data, showing that phage infection can drive MRSA populations to reduced virulence phenotypes.

## Combination treatment between phages and β-lactam

The aforementioned results suggest that MRSA cells evolve phage resistance following infection, which is associated with trade-offs in virulence and β-lactam resistance. We next asked how MRSA populations would evolve under co-treatment with phage and β-lactam. In principle, these two simultaneous selective pressures could drive the evolution of resistance against both the phage and antibiotic, negating the trade-offs in drug resistance. To test this, we performed checkerboard assays with phage and oxacillin on MRSA252, MW2, and LAC. Serial dilutions of Evo2 or ΦStaph1N were mixed with serial dilutions of oxacillin on a 96-well plate (*Figure 6A*), after which MRSA strains were added to the plate and allowed to grow for 24 hr. Following 24 hr, 1% of the culture in each well was transferred into a fresh plate well with nonselective media, and the cultures were allowed to grow for another 24 hr (48 hr total). Throughout the experiment, the cell density was monitored by measuring the optical density in each plate well.

We first examined how MRSA grew in combinations of Evo2 and antibiotic (*Figure 6A*, top row). For MRSA252, cells grew at low levels of phage (MOI of 0.01 or less) and oxacillin (<0.125 µg/mL; *Figure 6A*). LAC displayed greater sensitivity, showing no detectable growth after 48 hr in the presence of phage, irrespective of the presence of oxacillin. For MW2, cells showed limited growth at MOIs <1 and oxacillin levels <0.125 µg/mL. For each strain, we picked cells from wells with an $OD_{600}$ >0.5 and inoculated them into fresh, non-selective media. As a control, we also regrew cells that were treated with neither phage nor oxacillin (well B1). For MRSA252, two (E2 and F2) wells contained viable MRSA cells, forming turbid cultures; for MW2, six (D4, E2, E3, E5, F2, and F3) wells picked regrew (*Figure 6B*). We posit that cells in the failed cultures had reduced viability from the phage and antibiotic treatment. We tested these regrown cultures for their phage and oxacillin susceptibility. As expected, MRSA252 and MW2 cells from the B1 control wells were sensitive to Evo2, but resistant to oxacillin. In MRSA252, cells from E2 and F2 were resistant to Evo2 infection and exhibited a 1000-fold decrease in MIC against oxacillin. Similarly, in MW2, the six viable cultures exhibited Evo2 resistance and a 10- to 100-fold decrease in MIC. Altogether, these results match with those of MRSA that underwent single selection with Evo2.

We next analyzed how MRSA grew under ΦStaph1N/β-lactam combinations (*Figure 6A*, bottom row). Neither MRSA252 nor LAC could grow in any ΦStaph1N/oxacillin combination; MW2, by contrast, grew across under a wide range of ΦStaph1N/oxacillin combinations. When ΦStaph1N-infected MW2 was treated with high (16 µg/mL) levels of oxacillin, recovered cells (wells D11 and E11) showed no reduction in MIC against oxacillin (*Figure 6C*). However, the plaquing efficiency of ΦStaph1N was reduced by 3–6 orders of magnitude. Whole genome sequencing revealed that these cells evolved a unique set of mutations different from those seen in the single phage treatment conditions (*Table 4*).

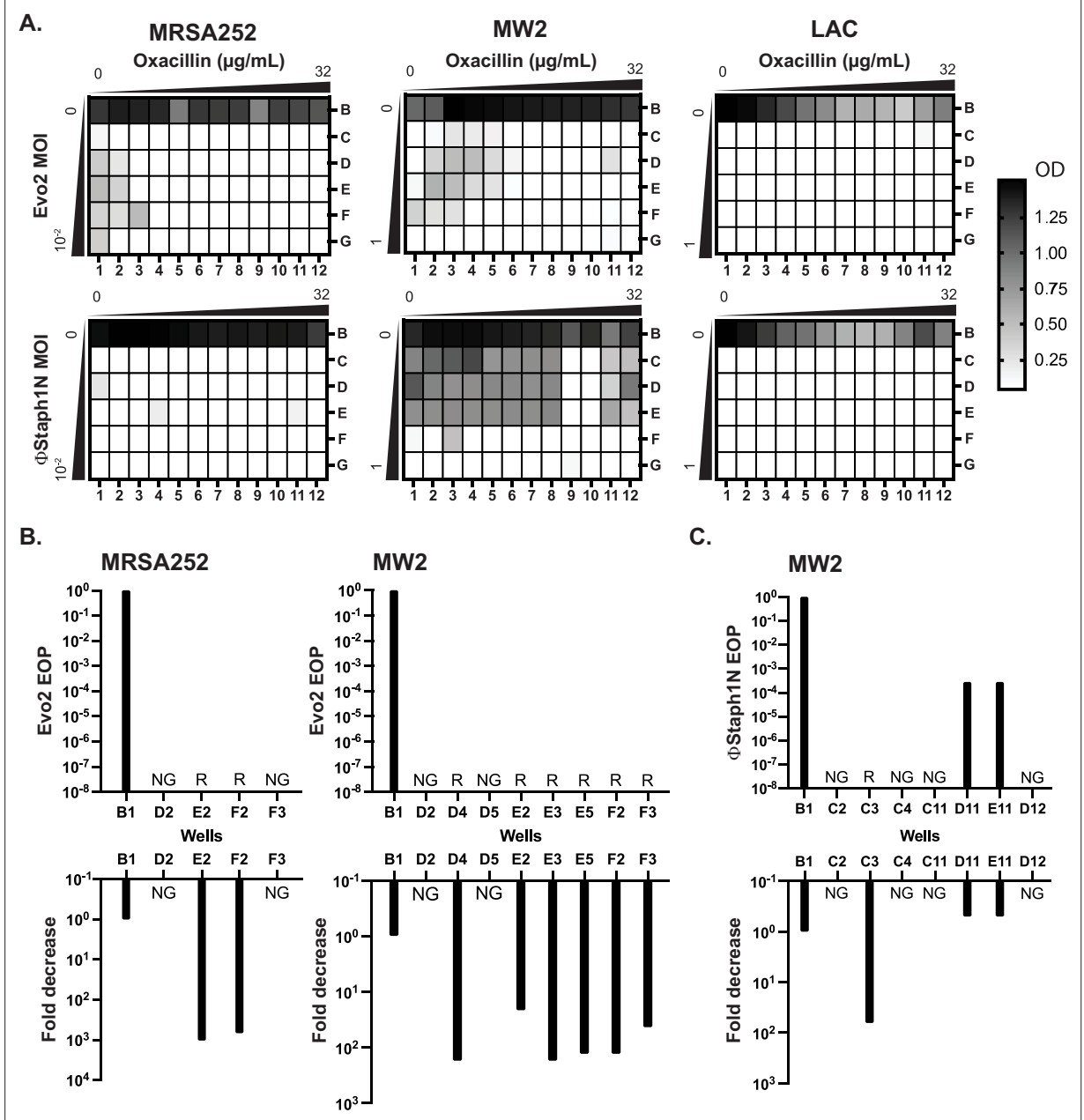

**Figure 6.** Co-treatment of MRSA with bacteriophage and β-lactam. (**A**) Checkerboard assays of MRSA strains with gradients of oxacillin and Evo2 (top panels) or ΦStaph1N (bottom panels). The oxacillin gradient is a twofold serial dilution of drug concentration (µg/mL), while the phage MOI gradient is a 10-fold serial dilution of MOI. The rows and columns of each plate are labeled with letters and numbers, respectively. The black-white gradient in each well reflects the optical density of the culture and is the mean value from three biological replicates. MRSA strains co-treated with oxacillin and (**B**) Evo2 or (**C**) ΦStaph1N were tested for their phage resistance and oxacillin resistance. The letter/number combination reflects the well from which the cells were picked for analysis. Wells that could not produce a viable culture are labeled as NG (no growth). For wells that regrew, we calculated the efficiency of plaquing (EOP) of phage and measured the fold reduction in oxacillin MIC. Cultures that showed no detectable viral plaques are labeled as resistant (**R**).

However, when ΦStaph1N-infected MW2 was co-treated with low levels (<0.125 µg/mL) of oxacillin, recovered cells (C3) displayed strong phage resistance and a 100-fold reduction in oxacillin resistance, mirroring phenotypes observed in the single phage treatment experiments.

We note that of the three MRSA strains tested, MW2 is the least sensitive to ΦStaph1N infection. We posit that the selective pressure exerted by high levels of β-lactam dominates over the selective pressure of ΦStaph1N, leading to the evolution of cells with continued β-lactam resistance and

limited phage resistance. However, at lower β-lactam levels, the pressure exerted by phage infection predominates, leading to the rise of cells with complete phage resistance and trade-off in β-lactam resistance. This dose-dependent selection by oxacillin is not observed when a more active phage (e.g. Evo2) is used. While limited to one MRSA strain, these results suggest that different phage/β-lactam combinations can produce divergent evolutionary outcomes in MRSA, each with potential clinical implications.

## Discussion

Here, we report the discovery that infection by certain phages can drive MRSA populations to evolve favorable genetic trade-offs between phage and β-lactam resistance. Exploiting genetic trade-offs has been proposed as a means to combat resistance in bacterial pathogens (*Oromí-Bosch et al., 2023*). Not only could phage treatments reduce the bacterial load of an infection, but they could also potentially resensitize bacterial populations to antibiotics against which they were previously resistant. We show that MRSA strains infected with *Kayviruses* ΦStaph1N, Evo2, and SATA8505 evolved resistance against phage, yet developed up to a 1000-fold loss in their MICs against β-lactam antibiotics. In addition, these evolved MRSA display reduced virulence phenotypes such as lower levels of hemolysis and clumping. Our findings show that phages can resensitize MRSA to β-lactams and even decrease their virulence, which are outcomes of significant biomedical value. However, not all staphylococcal phages can mediate these trade-offs: infection with ΦNM1γ6, a phage of the *Dubowvirus* genus, did generate phage resistance in MRSA, but did not produce a drop in β-lactam resistance. A major future direction will be to determine which types of phages elicit these beneficial evolutionary trade-offs in MRSA.

Our results also paint a complex picture of MRSA evolution during phage infection. Whole genome sequencing revealed that MRSA strains evolved distinct mutation profiles following phage infection, suggesting a multitude of evolutionary paths that different bacterial strains can undertake to evolve resistance against phage. Not only did MRSA strains evolve distinct mutations, but individual, phage-resistant clones accumulated multiple mutations in their genome. Nonetheless, different MRSA displayed a convergence of phenotypes in the form of phage resistance, reduced β-lactam resistance, and attenuated virulence. We posit that this convergence is caused by the involvement of the cell wall in all three phenotypic outcomes. Phages must interface with the cell wall, β-lactams target proteins associated with cell wall maintenance, and many *S. aureus* virulence factors are embedded within the cell wall or are secreted through it. Thus, any modifications to the integrity or chemical composition of the cell wall by phage resistance will impact β-lactam sensitivity and virulence. Cell wall maintenance is controlled by numerous genes, ranging from single proteins involved in cell wall synthesis, such as *femA*, to transcriptional regulators, such as *mgrA*, that control the expression of cell wall synthesis genes. Thus, the mutational patterns observed in each MRSA strain could reflect genetic solutions that enable the bacteria to adapt to the phage predation, while also maximizing the fitness for that particular strain.

Strikingly, MRSA heavily modulated transcriptional profiles following phage infection. We believe these altered expression profiles are a consequence of the genomic mutations that emerged in the various transcriptional regulators. We observed that evolved cells downregulated genes involved in quorum sensing, type VII secretion, and a variety of toxins. It is intriguing to speculate how the downregulation of these genes impacts MRSA interactions with other bacteria occupying the same ecological niche and with the host immune system. At the same time, MRSA strains also upregulated expression of select virulence factors, such as *spA*, which could represent 'trade-ups'. Trade-ups are thought of as non-selected traits that are enhanced following selection (e.g. cross-resistance between phage and antibiotic), which from a therapeutic perspective might be undesirable (*Kortright et al., 2021*). Future work will focus on assessing the risk of these trade-ups in light of the clinical benefit of reduced resistance and virulence.

Drug resistance in bacterial pathogens is an evolutionary problem and will require evolution-guided solutions to mitigate. Our findings highlight the ability of phages to dramatically alter the evolution and physiology of drug-resistant MRSA. Select phage treatments can force bacterial populations down evolutionary paths that make them vulnerable to antibiotics or the host immune system. Critically, this permits the re-deployment of agents that would otherwise remain ineffective, buying

time for new drug discoveries. We therefore hope that our work may suggest avenues of research into new phage-based treatment strategies against MRSA and other drug-resistant pathogens.

# Materials and methods

**Key resources table**

| Reagent type (species) or resource | Designation | Source or reference | Identifiers | Additional information |
|---|---|---|---|---|
| Strain, strain background (*Staphylococcus aureus*) | MRSA MW2 (USA400) | ***Baba et al., 2002*** | MW2 | |
| Strain, strain background (*Staphylococcus aureus*) | MRSA LAC (USA300) | ***Voyich et al., 2005*** | LAC | |
| Strain, strain background (*Staphylococcus aureus*) | MRSA252 (USA200) | ***Holden et al., 2004*** | MRSA252 | |
| Strain, strain background (*Staphylococcus epidermidis*) | *S. epidermidis* RP62a | ***Christensen et al., 1982*** | RP62a | methicillin-resistant biofilm-producing *S. epidermidis* |
| Strain, strain background (*Staphylococcus epidermidis*) | *S. epidermidis* LM1680 | ***Jiang et al., 2013*** | LM1680 | Derived from *S. epidermidis* RP62a; carries genomic deletion that inactivates biofilm production |
| Other | ΦStaph1N | ***Łobocka et al., 2012*** | ΦStaph1N | Bacteriophage of the *Kayvirus* genus |
| Other | Evo2 | This study | Evo2 | Derived from ΦStaph1N |
| Other | ΦNM1γ6 | Marraffini laboratory | ΦNM1γ6 | Bacteriophage of the *Dubowvirus* genus, lytic version of temperate phage ΦNM1 |
| Other | ΦNM4γ4 | Marraffini laboratory | ΦNM4γ4 | Bacteriophage of the *Dubowvirus* genus, lytic version of temperate phage ΦNM4 |
| Other | Φ12 | Marraffini laboratory | Φ12 | Bacteriophage of the *Triavirus* genus |
| Other | Andhra | Hatoum-Aslan laboratory | Andhra | Bacteriophage of the *Andravirus* genus, infects *S. epidermidis* |
| Other | SATA8505 | Environmental isolate; ***Pincus et al., 2015*** | SATA8505 | Bacteriophage of the *Kayvirus* genus, isolated from the environment in this study |
| Strain, strain background (*Staphylococcus aureus*) | ADL1 | Levin laboratory; ***Land et al., 2015*** | ADL1 | USA300, Patient isolate |
| Strain, strain background (*Staphylococcus aureus*) | ADL2 | Levin laboratory; ***Land et al., 2015*** | ADL2 | USA300, Patient isolate |
| Strain, strain background (*Staphylococcus aureus*) | ADL3 | Levin laboratory; ***Land et al., 2015*** | ADL3 | USA300, Patient isolate |
| Strain, strain background (*Staphylococcus aureus*) | ADL4 | Levin laboratory; ***Land et al., 2015*** | ADL4 | USA300, Patient isolate |
| Strain, strain background (*Staphylococcus aureus*) | ADL5 | Levin laboratory; ***Land et al., 2015*** | ADL5 | USA300, Patient isolate |
| Strain, strain background (*Staphylococcus aureus*) | ADL6 | Levin laboratory; ***Land et al., 2015*** | ADL6 | USA300, Patient isolate |
| Strain, strain background (*Staphylococcus aureus*) | ADL7 | Levin laboratory; ***Land et al., 2015*** | ADL7 | USA300, Patient isolate |
| Strain, strain background (*Staphylococcus aureus*) | ADL8 | Levin laboratory; ***Land et al., 2015*** | ADL8 | USA300, Patient isolate |
| Strain, strain background (*Staphylococcus aureus*) | ADL9 | Levin laboratory; ***Land et al., 2015*** | ADL9 | USA300, Patient isolate |

*Continued on next page*

*Continued*

| Reagent type (species) or resource | Designation | Source or reference | Identifiers | Additional information |
|---|---|---|---|---|
| Strain, strain background (*Staphylococcus aureus*) | ADL10 | Levin laboratory; *Land et al., 2015* | ADL10 | USA300, Patient isolate |
| Strain, strain background (*Staphylococcus aureus*) | ADL11 | Levin laboratory; *Land et al., 2015* | ADL11 | USA300, Patient isolate |
| Strain, strain background (*Staphylococcus aureus*) | ADL12 | Levin laboratory; *Land et al., 2015* | ADL12 | USA300, Patient isolate |
| Strain, strain background (*Staphylococcus aureus*) | ADL13 | Levin laboratory; *Land et al., 2015* | ADL13 | USA300, Patient isolate |
| Strain, strain background (*Staphylococcus aureus*) | ADL14 | Levin laboratory; *Land et al., 2015* | ADL14 | USA300, Patient isolate |
| Strain, strain background (*Staphylococcus aureus*) | ADL15 | Levin laboratory; *Land et al., 2015* | ADL15 | USA300, Patient isolate |
| Strain, strain background (*Staphylococcus aureus*) | ADL16 | *Land et al., 2015* | ADL16 | USA300, Patient isolate |
| Strain, strain background (*Staphylococcus aureus*) | ADL17 | Levin laboratory; *Land et al., 2015* | ADL17 | USA300, Patient isolate |
| Strain, strain background (*Staphylococcus aureus*) | ADL18 | Levin laboratory; *Land et al., 2015* | ADL18 | USA300, Patient isolate |
| Strain, strain background (*Staphylococcus aureus*) | ADL19 | Levin laboratory; *Land et al., 2015* | ADL19 | USA300, Patient isolate |
| Strain, strain background (*Staphylococcus aureus*) | ADL20 | Levin laboratory; *Land et al., 2015* | ADL20 | USA300, Patient isolate |
| Strain, strain background (*Staphylococcus aureus*) | ADL21 | Levin laboratory; *Land et al., 2015* | ADL21 | USA300, Patient isolate |
| Strain, strain background (*Staphylococcus aureus*) | ADL22 | Levin laboratory; *Land et al., 2015* | ADL22 | USA300, Patient isolate |
| Strain, strain background (*Staphylococcus aureus*) | ADL23 | Levin laboratory; *Land et al., 2015* | ADL23 | USA300, Patient isolate |
| Strain, strain background (*Staphylococcus aureus*) | ADL24 | Levin laboratory; *Land et al., 2015* | ADL24 | USA300, Patient isolate |
| Strain, strain background (*Staphylococcus aureus*) | ADL25 | Levin laboratory; *Land et al., 2015* | ADL25 | USA300, Patient isolate |
| Strain, strain background (*Staphylococcus aureus*) | ADL26 | Levin laboratory; *Land et al., 2015* | ADL26 | USA300, Patient isolate |
| Strain, strain background (*Staphylococcus aureus*) | ADL27 | Levin laboratory; *Land et al., 2015* | ADL27 | USA300, Patient isolate |
| Strain, strain background (*Staphylococcus aureus*) | ADL28 | Levin laboratory; *Land et al., 2015* | ADL28 | USA300, Patient isolate |
| Strain, strain background (*Staphylococcus aureus*) | ADL29 | Levin laboratory; *Land et al., 2015* | ADL29 | USA300, Patient isolate |
| Strain, strain background (*Staphylococcus aureus*) | ADL30 | Levin laboratory; *Land et al., 2015* | ADL30 | USA300, Patient isolate |
| Strain, strain background (*Staphylococcus aureus*) | AH1263 | Horswill laboratory | AH1263 | LAC, Erm[S] |
| Strain, strain background (*Staphylococcus aureus*) | AH3455 | Horswill laboratory | AH3455 | LAC *mgrA*::tetM |
| Strain, strain background (*Staphylococcus aureus*) | AH1975 | Horswill laboratory | AH1975 | LAC Δ*arl* |

*Continued on next page*

*Continued*

| Reagent type (species) or resource | Designation | Source or reference | Identifiers | Additional information |
|---|---|---|---|---|
| Strain, strain background (*Staphylococcus aureus*) | AH1525 | Horswill laboratory | AH1525 | LAC *sarA*::kan |
| Strain, strain background (*Staphylococcus aureus*) | AH843 | Horswill laboratory | AH843 | MW2 |
| Strain, strain background (*Staphylococcus aureus*) | AH3456 | Horswill laboratory | AH3456 | MW2 *mgrA*::tetM |
| Strain, strain background (*Staphylococcus aureus*) | AH3060 | Horswill laboratory | AH3060 | MW2 *arl*::tet |
| Strain, strain background (*Staphylococcus aureus*) | AH5679 | Horswill laboratory | AH5679 | MW2 *sarA*::Tn(Erm) |
| Software, algorithm | Filtlong | *Wick, 2021* | v.0.2.1; RRID:SCR_024020 | |
| Software, algorithm | Minimap2 | *Li, 2018* | v.2.22; RRID:SCR_018550 | |
| Software, algorithm | SAMtools | *Li et al., 2009* | v.13; RRID:SCR_002105 | |
| Software, algorithm | Bakta | *Schwengers et al., 2021* | v.1.10.3; RRID:SCR_026400 | |
| Software, algorithm | eggNOG-mapper | *Cantalapiedra et al., 2021* | v.2.1.12; RRID:SCR_021165 | |
| Software, algorithm | Bowtie 2 | *Langmead and Salzberg, 2012* | v2.5.4; RRID:SCR_016368 | |
| Software, algorithm | featureCounts | *Liao et al., 2014* | v.2.0.8; RRID:SCR_012919 | |
| Software, algorithm | R Project for Statistical Computing | https://www.r-project.org/ | v.4.4.0; RRID:SCR_001905 | |
| Software, algorithm | DESeq2 | *Love et al., 2014* | v.1.44.0; RRID:SCR_015687 | |
| Software, algorithm | tidyverse | *Wickham et al., 2019* | v.2.0.0; RRID:SCR_019186 | |
| Software, algorithm | EnhancedVolcano | https://github.com/kevinblighe/EnhancedVolcano | v.1.22.0; RRID:SCR_018931 | |
| Software, algorithm | Trimmomatic | *Bolger et al., 2014* | v.0.39; RRID:SCR_011848 | |
| Software, algorithm | SPAdes | *Bankevich et al., 2012* | v.4.0.0; RRID:SCR_000131 | |
| Software, algorithm | blastn | *Altschul et al., 1997* | v2.16.0; RRID:SCR_001598 | |
| Software, algorithm | checkv | *Nayfach et al., 2021* | v.1.0.3 | |
| Software, algorithm | taxmyphage | *Millard et al., 2025* | v.0.3.4 | |
| Software, algorithm | Flye | *Kolmogorov et al., 2019* | v.2.9.3; RRID:SCR_017016 | |
| Software, algorithm | Prodigal | *Hyatt et al., 2010* | v.2.6.3; RRID:SCR_011936 | |

## Strains and culture conditions

The bacterial strains used in this study are listed in *Supplementary file 1, table S1*. Unless otherwise indicated, all MRSA strains were grown in Brain Heart Infusion (BHI) media at 37 °C with shaking (235 RPM).

## Plate-based plaque assay

Bacterial lawns were prepared by mixing 100 µL of an overnight culture with 5 mL of melted BHI agarose (top agar). The bacteria and top agar mixture were poured onto a solid BHI plate. The plate was dried for 10 min. 10-fold serial dilutions ($10^0$ -$10^{-7}$ unless otherwise noted) of phage were then spotted on the bacterial lawn. Plates were then incubated at 37 °C for 16 hr. Phage titer in plaque-forming units per µL (pfu/µL) was then calculated.

## Phage infection assay

MRSA strains were plated onto BHI agar plates and grown overnight. Individual colonies from the parental strains (also referred to as P0) were picked. Single colonies were inoculated in a round bottle

tube containing 5 mL BHI broth. The cultures were incubated at 37 °C, 235 RPM for 24 hr. The grown P0 cultures were then diluted 1:100 into fresh 5 mL BHI broth. At the early log phase (OD ~0.3), the bacterial cultures were treated with phage at an MOI of 0.1, unless indicated otherwise. The treated bacterial cultures were incubated at 37 °C with shaking (235 RPM) for 24 hr. The cultures were then passaged 1:100 into fresh 5 mL BHI broth. This passage was then grown at 37 °C, 235 RPM, for another 24 hr. Surviving cultures were then used for both phenotypic assays and sequencing experiments. As a negative control, MRSA strains were passaged using the steps described above without phage treatment (mock).

## MIC assay

Bacterial lawns were normalized to contain $1 \times 10^8$ CFU/mL bacteria mixed with top agar for a total volume of 5 mL. The bacteria and top agar mixture is poured onto a solid BHI plate. The plate was dried for 10 min. MIC with increasing concentrations of antibiotics was placed on the semi-dried bacterial lawn and allowed to dry for 10 min. The plates are then incubated at 37 °C overnight. For analysis, the plates were imaged, and the MIC of the bacteria was determined. The MIC is determined at the edge of the inhibition ellipse that intersects the side of the strip.

## Rabbit blood hemolysis

Phage-treated or mock-treated cultures were diluted to an OD600 of 0.1. 5 µL of this dilution was spotted on rabbit blood TSB agar plates and incubated at 37 °C for 24 hr. The area of clearance was determined by the following formula:

[π (diameter of clearance/2) (*Tomasz, 1979*)] - [π(diameter of bacterial spot/2) (*Tomasz, 1979*)].

## Clumping assay

Clumping assays were performed as described previously (*Crosby et al., 2021*). In short, overnight cultures were diluted 1:100 and incubated at 37 until the cultures reached an $OD_{600}$ of 1.5. At this point, 1.5 mL of culture was washed two times and resuspended with PBS. Lyophilized human plasma was added for a final concentration of 1.25%. Resuspended cells were left to sit statically at room temperature. 100 µL were taken from the top of the cell suspensions in 30-min intervals, and the $OD_{600}$ was measured.

## Biofilm assay

Biofilm assay was performed using the crystal violet method as outlined (*Feoktistova et al., 2016*). In brief, overnight cultures grown in BHI at 37 °C were back diluted 1:100 into a 96-well bottomed microwell plate. The plates were incubated without shaking at 37 °C overnight. The contents in the plate were discarded and washed with PBS. Biofilm fixation was done with sodium acetate (2%). Crystal violet (0.1%) was used for staining, followed by a final wash with PBS. Absorbance at 600 nm was read using a spectrophotometer.

## DNA sequencing and genome assembly

Following published protocols, genomic DNA from bacteria and phage was isolated using phenol-chloroform extraction. Purified DNA was sent to Plasmidsaurus and SeqCenter for Nanopore and Illumina sequencing, respectively. Reference genomes for bacterial strains were assembled using Flye v2.9.3 with default settings for long reads. This resulted in 1 singular contig assembly for 252 (2902592bp, 125 x) and MW2 (2820460, 600 x), and 3 contigs for LAC (2907712, 645 x). Phage assemblies for Evo2 and ΦStaph1N were done with the SPADES assembler v3.15.5.

Open-reading frames (ORFs) were called on the assembled bacterial genomes using Prodigal v.2.6.3 (*Hyatt et al., 2010*), resulting in a gene-feature file (GFF) and translated genes as.faa and .fna formats. We used BLASTp (Accessed June 4th, 2024), against the protein BLAST database swissprot_2023-06-28, with an expectation value cutoff of 0.001. The top hit for each ORF was used as the final functional annotation. Additionally, we annotate the genomes using Bakta v.1.10.3.

## Mutation identification

A total of 27 genomic samples were collected. DNA was extracted and sent for long-read sequencing using Oxford Nanopore Technology (Plasmidsaurus, San Francisco, USA). Reads were filtered using

filtlong v0.2.1 using default settings (https://github.com/rrwick/Filtlong, *Wick, 2021*) and with the -p flag 95 (keeping 95% of the best reads). Quality-filtered long reads were mapped against the respective genomes using minimap2 2.22-r1101 (*Li, 2018*), resulting in 1 alignment file output per sample (.sam file). The read mapping software Minimap2 (*Li, 2018*) was selected because of its suitability to map long reads. Samtools v1.20 (*Li et al., 2009*) was used to convert the .sam files into .bam files, sort the bam file, index the bam files, and generate a coverage table for each position along the alignment.

Reference files (fasta and GFF files) and the 53 'sorted.bam' alignment files were imported into Geneious. Variant calling was performed using Geneious Prime Geneious Prime 2024.0.2 (https://www.geneious.com/), using the custom settings: 10% coverage, minimum 95% variant frequency threshold, and the option for 'Analyze effect of variants on translation' checked. Variant results and genome annotations table files were exported as a tab-separated table and visualized using R v4.4.0, mostly with the tidyverse package. Sequencing data processing, quality filtering, and mapping were performed at the Center for High-Throughput Computing (https://chtc.cs.wisc.edu/). BLAStp was performed on usegalaxy.eu (Accessed June 4th, 2024).

## RNA purification

Parental and evolved MRSA strains were diluted 1:100 in BHI broth and incubated at 37 °C, 235 RPM until they reached an $OD_{600}$ of 1.5. 500 µL of culture were transferred into a microcentrifuge and 1 mL of RNAprotect Bacteria reagent (QIAGEN) was added. The mixture was vortexed for 5 s and incubated at room temperature for 5 min. The tubes were centrifuged for 10 mins at $5000 \times g$, and the supernatant discarded. Bacterial pellets were resuspended in 80 µL of phosphate buffered saline and 10 µL of lysostaphin solution (1 mg/mL stock). The suspensions were incubated at 37 °C, with shaking, until the solution looked clear (~30 minutes). 10 µL of 10% sarkosyl was then added and the tube mixed, after which 300 µL of TRIzol reagent (Invitrogen) was added. RNA purification was performed following the protocol from the Direct-zol RNA Miniprep Plus kit (ZYMO Research). DNase I treatment was performed as recommended, and the samples were eluted in 75 µL of DNase/RNase-free water.

## RNA sequencing and differential gene expression analysis

Purified RNA was prepared and sequenced on an Illumina sequencing platform at the UW-Madison Gene Expression Center. RNA-seq data was collected in the parental and evolved MRSA strains to assess differentially expressed genes. Cleaned RNA reads were mapped onto the LAC and MW2 reference genomes using bowtie2 v2.5.4 (*Langmead and Salzberg, 2012*), and featureCount v2.0.8 (*Liao et al., 2014*) from the software subreads was used to generate a read count matrix. Two read count matrices (one for LAC and one for MW2) were imported into R v.4.4.0 for processing with DESeq2 (version 1.44.0; *Love et al., 2014*). Figures were generated using the packages tidyverse (version 2.0.0) and EnhancedVolcanoPlots (version 1.22.0). To generate the volcano plots, we chose an adjusted p-value of 0.002 and a $\log_2$ fold change ($\log_2$FC) of $< -2$ or $>2$. Multiple 'unknown' genes were deemed significant (adjusted p-value <0.002, abs($\log_2$FC) $\geq$ 2) in both LAC and MW2. To compare the results between the genomes, we performed a protein clustering analysis using MMseqs2 version b804fbe384e6f6c9fe96322ec0e92d48bccd0a42 between all the Bakta-generated amino acid sequences (.faa files) from LF and MW2 (*Steinegger and Söding, 2017*). Then we considered any protein sharing over 80% identity to be the 'same' to generate a summary table showing up-regulation and down-regulation among the two genomes.

## Checkerboard assay for phage-antibiotic synergy

The overnight cultures of MRSA252, Lac-Fitz, and MW2 were back-diluted 1:100 in BHI and incubated at 37 °C until the culture reached mid-log phase. The culture was then inoculated into each well of the 96-well plate containing a gradient of oxacillin and phage (ΦStaph1N or Evo2). The oxacillin gradient was a twofold serial dilution, while the phage MOI gradient was a 10-fold serial dilution. The plates were then placed at 37 °C with shaking (235 RPM) for 24 hr.

Following 24 hr, each well from the plate was then passaged 1:100 into another 96-well plate with fresh BHI and grown at 37 °C with shaking (235 RPM) for another 24 hr. Following 48 hr of growth, the $OD_{600}$ was measured with a plate reader. Each checkerboard assay was performed in three biological replicates. A selection of surviving cells from individual wells (cells with an $OD_{600}$ reading of >0.5) were then picked for phage plaquing and MIC assays. Efficiency of plaquing (EOP) was calculated by

the following formula: plaque titer of treated cells/plaque titer of non-treated cells. To determine the mutation profile of survivor cells, genomic extraction, Illumina sequencing, and mutational analysis described above were used.

## Language accessibility

A translation of the manuscript into Spanish can be found in *Supplementary file 3*, titled "Infección por bacteriófagos provoca la pérdida de resistencia a los β-lactámicos en *Staphylococcus aureus* resistente a la meticilina."

## Acknowledgements

MT is supported by the SciMed GRS program at UW-Madison; AJHV is supported by the NSF Graduate Research Fellowship Program; CYM is supported by start-up funds from the Department of Bacteriology at UW-Madison and the Margeret Q Landenberger Foundation. We are grateful to Dr. Wilmara Salgado-Pabón, Dr. Petra Levin, and Dr. Alexander Horswill for providing us with MRSA strains and helpful suggestions. We thank the Marraffini and Hatoum-Aslan laboratories for providing us with bacteriophage samples. Computational analyses were performed using the UW-Madison Center for High Throughput Computing. We also thank all members of the Mo lab for their scientific input.

## Additional information

### Competing interests

My Tran, Angel J Hernandez Viera, Charlie Y Mo: Filed a provisional patent for the work presented in this paper through the Wisconsin Alumni Research Foundation (P240267US01). The other authors declare that no competing interests exist.

### Funding

| Funder | Grant reference number | Author |
| --- | --- | --- |
| University of Wisconsin | SciMed Graduate Research Scholars | My Tran |
| National Science Foundation | Graduate Research Fellowship Program | Angel J Hernandez Viera |
| Margaret Q. Landenberger Research Foundation | Margaret Q. Landenberger Foundation Award | Charlie Y Mo |
| Department of Bacteriology at UW-Madison | | Charlie Y Mo |

The funders had no role in study design, data collection and interpretation, or the decision to submit the work for publication.

### Author contributions

My Tran, Angel J Hernandez Viera, Conceptualization, Formal analysis, Investigation, Methodology, Writing – original draft, Writing – review and editing; Patricia Q Tran, Data curation, Software, Formal analysis, Writing – review and editing; Erick D Nilsen, Lily Tran, Investigation; Charlie Y Mo, Conceptualization, Data curation, Formal analysis, Supervision, Funding acquisition, Investigation, Methodology, Writing – original draft, Project administration, Writing – review and editing

### Author ORCIDs

Angel J Hernandez Viera ⓘ https://orcid.org/0009-0008-4439-3308
Patricia Q Tran ⓘ https://orcid.org/0000-0003-3948-3938
Charlie Y Mo ⓘ https://orcid.org/0009-0000-3535-6734

Reviewer #1 (Public review): https://doi.org/10.7554/eLife.102743.3.sa1
Reviewer #2 (Public review): https://doi.org/10.7554/eLife.102743.3.sa2

Author response https://doi.org/10.7554/eLife.102743.3.sa3

## Additional files

### Supplementary files

Supplementary file 1. List of mutations found in phage-treated and untreated *S. aureus* strains.

Supplementary file 2. List of significantly up-or downregulated transcripts in MRSA strains MW2 and LAC.

Supplementary file 3. Spanish translation of this article.

MDAR checklist

### Data availability

Sequences have been deposited to NCBI under the BioProject ID PRJNA1263016. Code has been deposited to https://github.com/MolabUW/mrsa-project-eLife2025/ (copy archived at *Tran, 2025*). All data generated or analyzed during this study are included in the manuscript and supporting files; source data files have been uploaded for Figures 1, 2, 3, 5, and their corresponding figure supplements.

The following dataset was generated:

| Author(s) | Year | Dataset title | Dataset URL | Database and Identifier |
|---|---|---|---|---|
| Tran M, Hernandez Viera AJ, Tran PQ, Nilsen E, Tran L, Cy Mo | 2025 | Sequencing data of MRSA strains subjected to bacteriophage infections | https://www.ncbi.nlm.nih.gov/bioproject/PRJNA1263016 | NCBI BioProject, PRJNA1263016 |

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
