## [Editor Report · eLife Assessment]

The manuscript explores how bacterial evolution in the presence of lytic phages modulates b-lactams resistance and virulence properties in methicillin-resistant *Staphylococcus aureus* (MRSA). This **important** work improves our knowledge of how mutation in genes required for phage infection confers sensitivity to b-lactams and alter virulence properties. Altogether, the findings are **convincing**.

---

## [Referee Report · Reviewer #1 (Public review)]

Summary:

These authors have asked how lytic phage predation impacts antibiotic resistance and virulence phenotypes in methicillin-resistant *Staphylococcus aureus* (MRSA). They report that staphylococcal phages cause MRSA strains to become sensitized to b-lactams and to display reduced virulence. Moreover, they identify mutations in a set of genes required for phage infection that may impact antibiotic resistance and virulence phenotypes.

Strengths:

Phage-mediated re-sensitization to antibiotics has been reported previously but the underlying mutational analyses have not been described. These studies suggest that phages and antibiotics may target similar pathways in bacteria.

Weaknesses:

One limitation is the lack of mechanistic investigations linking particular mutations to the phenotypes reported here. This limits the impact of the work.

Another limitation of this work is the use of lab strains and a single pair of phages. However, while incorporation of clinical isolates would increase the translational relevance of this work it is unlikely to change the conclusions.

Comments on revisions:

The authors have addressed my concerns.

---

## [Referee Report · Reviewer #2 (Public review)]

Summary:

The work presented in the manuscript by Tran et al deals with bacterial evolution in the presence of bacteriophage. Here, authors have taken three methicillin-resistant *S. aureus* strains that are also resistant to beta-lactams. Eventually, upon being exposed to phage, these strains develop beta-lactam sensitivity. Besides this, the strains also show other changes in their phenotype such as reduced binding to fibrinogen and hemolysis.

Strengths:

The experiments carried out are convincing to suggest such in vitro development of sensitivity to the antibiotics. Authors were also able to "evolve" phage in similar fashion thus showing enhanced virulence against the bacterium. In the end, authors carry out DNA sequencing of both evolved bacteria and phage and show mutations occurring in various genes. Overall, the experiments that have been carried out are convincing.

Weaknesses:

None. In the current version of the manuscript, I find the study complete.

---

## [Author Response]

The following is the authors’ response to the original reviews:

We sincerely thank the reviewers for their thoughtful review and feedback. We believe that our work will provide valuable insights into how MRSA evolves under bacteriophage predation and stimulate efforts to use genetic trade-offs to combat drug resistance. We have substantially revised the paper and performed several additional experiments to address the reviewers' questions and concerns.

Summary:

(1) Testing for genetic trade-offs in additional *S. aureus* strains

We obtained 30 clinical isolates of the *S. aureus* USA300 strain that were isolated between 2008 and 2011 (see Table S1). We first tested the FStaph1N, Evo2, and FNM1g6 phages against this expanded strain panel and found that Evo2 showed strong activity against all 30 strains (Table S4). We tested whether Evo2 infection could elicit trade-offs in b-lactam resistance for a subset of these strains. We found that Evo2 infection caused a ~10-100-fold reduction in their MIC against oxacillin. This data is now incorporated into a revised Figure 2 in panel C.

(2) Testing additional staphylococcal phages

We isolated from the environment a phage called SATA8505. Similar to FStaph1N and Evo2, SATA8505 belongs to the *Kayvirus* genus and infects the MRSA strains MRSA252, MW2, and LAC. Phage-resistant MRSA recovered following SATA8505 infection also showed a strong reduction in oxacillin resistance (Figure S5). Furthermore, we confirmed that resistance against FNM1g6, which belongs to the *Dubowvirus* genes, does not elicit tradeoffs in b-lactam resistance (Figure S4). Sequencing analysis of FNM1g6 - resistant LAC strains showed a different mutation *fmhC*, which was not observed with the FStaph1N and Evo2 phages (Table 1). We have added this new data into the main text and supplemental figures and tables. Future work will focus on obtaining comprehensive analysis of a wide range of phage families.

(3) Testing additional antibiotics

We also expanded our trade-off analysis include wider range of antibiotic classes (Table S3). Overall, the loss of resistance appears to be confined to b-lactams.

(4) Genetic analysis of ORF141

In order determine the function of ORF141, which is mutated in Evo2, we attempted to clone wild-type ORF141 into a staphylococcal plasmid and perform complementation assays with Evo2. Unfortunately, obtaining the plasmid-borne wild-type ORF141 has proven to be tricky, as all clones developed frameshift or deletions in the open reading frame. We posit that the gene product of ORF141 is toxic to the bacteria. We are currently working on placing the gene under more stringent expression conditions but feel that these efforts fall outside of the scope of this paper.

(5) Testing the effect of single mutants

Our genomic analysis showed that phage-resistant MRSA evolved multiple mutations following phage infection, making it difficult to determine the mechanism of each mutation alone. For example, phage-resistant MW2 and LAC evolved nonsense mutations in transcriptional regulators *mgrA*, *arlR*, and *sarA*. To test whether these mutations alone were sufficient to confer resistance, we obtained MRSA strains with single-gene knockouts of *mgrA*, *arlR*, and *sarA* and tested their ability to resist phage. We observed that deletion of *mgrA* in the MW2 resulted in a modest reduction in phage sensitivity (Figure S7). However, we did not the observe any changes in the other mutant strains. These results suggest that phage resistance in these strains is likely caused by a combination of mutations. Determining the mechanisms of these mutations is the focus if our future work.

(6) Transcriptomics of phage-resistant MRSA strains

To further assess the effects of the phage resistance mutations, we performed bulk RNA-seq on phage-resistant MW2 and LAC strains and compared their differential expression levels to the respective wild-type strains. We picked these strains because our genomic data showed that they had evolved mutations in known transcriptional regulators (e.g. *mgrA*). Our analysis shows that both strains significantly modulate their gene expression (Figure 4). Notably, both strains upregulate the cell wall-associated protein *ebh*, while downregulating several genes involved in quorum sensing, virulence, and secretion. We have included this new data in Figure 4 and Table S5 and added an entire section in the manuscript discussing these results and their implications.

(7) Co-treatment of MRSA with phage and b-lactam

We performed checkerboard experiments on MRSA strains with phage and b-lactam gradients (Figure 6). We found that under most conditions, MRSA cells were only able to recover under low phage and b-lactam concentrations. Notably, these recovered cells were still phage resistant and b-lactam sensitive. However, under one condition where MW2 was treated with FStaph1N and b-lactam, we found that some recovered cells still had high levels of b-lactam resistance, showing a distinct mutational profile. We discuss these results in detail in the main text.

**Reviewer # 1:**
Strengths:Phage-mediated re-sensitization to antibiotics has been reported previously but the underlying mutational analyses have not been described. These studies suggest that phages and antibiotics may target similar pathways in bacteria.

We thank Reviewer 1 for this assessment. We hope that the data provided in this work will help stimulate further inquiries into this area and help in the development of better phage-based therapies to combat MRSA.

Weaknesses:One limitation is the lack of mechanistic investigations linking particular mutations to the phenotypes reported here. This limits the impact of the work.

We acknowledge the limitations of our initial analysis. We note (and cite) that separate studies have already linked mutations in *femA*, *mgrA*, *arlR*, and *sarA* with reduced b-lactam resistance and virulence phenotypes in MRSA, but not to phage resistance. For the other mutations, we could not find literature linking them to our observed phenotypes. We analyzed the effects of single gene knockouts of *mgrA*, *arlR*, and *sarA* on MRSA’s phage resistance. However, as shown above, the results only showed modest effects on phage resistance in the MW2 strain (see Figure S7 and lines 309-317). We therefore believe that mutations in single genes are not sufficient to cause the trade-offs in phage/ b-lactam resistance. Because each MRSA strain evolved multiple mutations (e.g. MW2 evolved 6 or more mutations), we feel that determining the effects of all possible permutations of those mutations was beyond the scope of the paper.

However, to bridge the mutational data with our phenotypic observations, we performed RNAseq and compared the transcriptomes of un-treated and phage-treated MRSA strains (see Figure 4, Table S5, and lines 337-391). Our results show that phage-treated MRSA strains significantly modulate their transcript levels. Indeed, some of the changes in gene expression can explain for the phenotypic observations (e.g. overexpression of *ebh* can lead to reduced clumping). Further, the results shown some unexpected patterns, such as the downregulation of quorum sensing genes or genes involved in type VII secretion.

Another limitation of this work is the use of lab strains and a single pair of phages. However, while incorporation of clinical isolates would increase the translational relevance of this work it is unlikely to change the conclusions.

We thank the reviewer for this suggestion. We would like to clarify that MW2, MRSA252, and LAC are pathogenic clinical isolates that were isolated between 1997 and 2000’s. However, we acknowledge that, because these 3 strains have been propagated for many generations, they might have acquired laboratory adaptations. We therefore obtained 30 USA300 clinical strains that were isolated in more recent years (~2008-2011) and tested our phages against them. We note that these clinical isolates (generously provided by Dr. Petra Levin’s lab) were preserved with minimal passaging to reduce the effects of laboratory adaptation. We found that the Evo2 phage was able to elicit oxacillin trade-offs in those strains as well. (see Table S1, Table S7, Fig 2C, and lines 210 – 225)

For the phages, we had to work with phage(s) that could infect all three MRSA strains. That is why in our initial tests, we focused on FStaph1N and Evo2, both members of the *Kayvirus* genus. Now in our revised work, we extend our analysis to FNM1g6, a member of the Dubowvirus genus, that also infects the LAC strain, but not MW2 and MRSA252. We find that FNM1g6 is unable to drive trade-offs in b-lactam resistance (see lines 229 – 238). Next, we analyzed the effects of SATA8505, also a member of the Kayvirus genus. Here, we observed that SATA8505 can elicit trade-offs in b-lactam resistance (see Figure S5 and lines 238 – 246). These results suggest that not all staphylococcal phages can elicit these trade-offs and call for more comprehensive analyses of different types of phages.

**Reviewer #1 (Recommendations for the authors):**
Specific questions:(1) The Evo2 isolate is an evolved version of phage Staph1N with more potent lytic activity. Is this reflected in more pronounced antibiotic sensitivity?

We did not observe that Evo2-treated MRSA cells showed more sensitivity towards b-lactams. However, we did observe that Evo2 was able to elicit these trade-offs at lower multiplicities of infection (MOI) (see lines 173 – 176 and Figure S2). Further, we did observe that Evo2 caused a greater trade-off in virulence phenotypes (hemolysis and cell agglutination) (see lines 416 - 419 lines 433 – 435, and Figure 5)

In our revisions, we also tested Evo2-treated MRSA against a wide range of antibiotics. We did not observe significant changes in MICs against those agents.

(2) Are there mutations in the SCCmec cassette or the MecA gene after selection against ΦStaph1N?

We did not observe any mutations in known resistance genes SCCmec or *blaZ*. Furthermore, we did not see any differential expression of those genes in our transcriptomic data (see lines 344 and 346).

(3) The authors report that phage ΦNM1γ6 does not induce antibiotic sensitivity changes despite being effective against bacterial strain LAC. Were mutational sequencing studies performed with the resistant isolates that emerged against this strain? Can the authors hypothesize why these did not impact the virulence or resistance of LAC despite effective killing? How does this align with their models for ΦStaph1N?

We thank the reviewer for that insightful question. In our revised manuscript, we found that ΦNM1γ6 elicits a point mutation in the *fmhC* gene, which is involved in cell wall maintenance (see lines 326 – 335). To our knowledge, this point mutation has not been linked to phage resistance or drug sensitivity MRSA. Notably this mutation was not observed with ΦStaph1N or Evo2. We therefore speculate that ΦNM1γ6 binds to a different receptor molecule on the MRSA cell wall.

(4) If I understand correctly, the authors attribute these effects of phage predation on antibiotic sensitivity and virulence to orthogonal selection pressures. A good test of this model would be to examine the mutations that emerge in antibiotic/phage co-treatment. This should be done.

We thank the reviewer for this suggestion. As described in the summary section above, we performed checkerboard experiments on MRSA strains with phage and b-lactam gradients (see lines 440 – 494 and Figure 6). We found that under most conditions, MRSA cells were only able to recover under low phage and b-lactam concentrations. Notably, these recovered cells were still phage resistant and b-lactam sensitive. However, under one condition where MW2 was treated with FStaph1N and b-lactam, we found that some recovered cells still had high levels of b-lactam resistance and only limited phage resistance, showing a distinct mutational profile (Figure S6). Under these conditions, we think that the selective pressure exerted by FStaph1N is “overcome” by the selective pressure of the high oxacillin concentration, a point that we discuss in the main text.

**Reviewer #2 (Public review):**
Summary:The work presented in the manuscript by Tran et al deals with bacterial evolution in the presence of bacteriophage. Here, the authors have taken three methicillin-resistant *S. aureus* strains that are also resistant to beta-lactams. Eventually, upon being exposed to phage, these strains develop beta-lactam sensitivity. Besides this, the strains also show other changes in their phenotype such as reduced binding to fibrinogen and hemolysis.Strengths:The experiments carried out are convincing to suggest such in vitro development of sensitivity to the antibiotics. Authors were also able to "evolve" phage in a similar fashion thus showing enhanced virulence against the bacterium. In the end, authors carry out DNA sequencing of both evolved bacteria and phage and show mutations occurring in various genes. Overall, the experiments that have been carried out are convincing.

We thank Reviewer 2 for their positive comments.

Weaknesses:Although more experiments are not needed, additional experiments could add more information. For example, the phage gene showing the HTH motif could be reintroduced in the bacterial genome and such a strain can then be assayed with wildtype phage infection to see enhanced virulence as suggested. At least one such experiment proves the discoveries regarding the identification of mutations and their outcome.

We thank the reviewer for this suggestion. We attempted to clone ORF141 into an expression plasmid and perform complementation experiments with Evo2 phage; however, all transformants that were isolated had premature stop-codons and frameshifts in the wild-type ORF141 insert that would disrupt protein function. We therefore think that the gene product of ORF141 might be toxic to the cells. We are currently working on placing the gene under more stringent transcriptional control but feel that these efforts fall outside of the scope of this paper.

Secondly, I also feel that authors looked for beta-lactam sensitivity and they found it. I am sure that if they look for rifampicin resistance in these strains, they will find that too. In this case, I cannot say that the evolution was directed to beta-lactam sensitivity; this is perhaps just one trait that was observed. This is the only weakness I find in the work. Nevertheless, I find the experiments convincing enough; more experiments only add value to the work.

We thank the reviewer for their comments. Because both phages and β-lactams interface with the bacterial cell wall, we posited that phage resistance would reduce resistance in cell wall targeting antibiotics. In our revisions, we have expanded our analysis to include a much wider range of antibiotic classes, including rifampicin, mupirocin, erythromycin, and other cell wall disruptors, such as daptomycin and teicoplanin. We did not observe any significant changes to the MICs of these other antibiotics (see Table S3 and lines 191-199). It therefore appears that the effects of these trade-offs are confined to beta-lactams.